# PAPER: Privacy-Preserving ResNet Models using Low-Degree Polynomial Approximations and Structural Optimizations on Leveled FHE

## Abstract

Recent work has made *non-interactive privacy-preserving inference* more practical by running deep Convolution Neural Network (CNN) with Fully Homomorphic Encryption (FHE). However, these methods remain limited by their reliance on *bootstrapping*, a costly FHE operation applied across multiple layers, severely slowing inference. They also depend on *high-degree polynomial approximations* of non-linear activations, which increase multiplicative depth and reduce accuracy by 2–5% compared to plaintext ReLU models. In this work, we focus on ResNets, a widely adopted benchmark architecture in privacy-preserving inference, and close the accuracy gap between their FHE-based non-interactive models and plaintext counterparts, while also achieving faster inference than existing methods. We use a *quadratic polynomial approximation* of ReLU, which achieves the theoretical minimum multiplicative depth for non-linear activations, along with a penalty-based training strategy. We further introduce *structural optimizations* such as node fusing, weight redistribution, and tower reuse. These optimizations reduce the required FHE levels in CNNs by nearly a factor of five compared to prior work, allowing us to *run ResNet models under leveled FHE without bootstrapping*. To further accelerate inference and recover accuracy typically lost with polynomial approximations, we introduce parameter clustering along with a joint strategy of data encoding layout and ensemble techniques. Experiments with ResNet-18, ResNet-20, and ResNet-32 on CIFAR-10 and CIFAR-100 show that our approach achieves up to $4\times$ faster private inference than prior work with comparable accuracy to plaintext ReLU models.

## 1 Introduction

*Machine Learning as a Service* (MLaaS) is increasingly adopted across industries as it provides access to powerful models without the need for in-house development or maintenance (Ribeiro et al., 2015). However, it raises serious privacy concerns since models are often trained on proprietary or sensitive data (Anwar et al., 2018; Özbayoglu et al., 2020), and inference requires clients to share private information (e.g., medical or financial records) with external service providers. *Privacy-Preserving Machine Learning* (PPML) addresses these risks using cryptographic methods, mainly secure *Multi-Party Computation* (MPC) (Zhou et al., 2024) and *Fully Homomorphic Encryption* (FHE) (Podschwadt et al., 2022), which protect both client data and provider models. Broadly, PPML methods fall into *interactive* and *non-interactive* categories (Lee et al., 2022). Interactive PPML, based on MPC, requires the client and the service provider to jointly perform inference (Liu et al., 2017; Juvekar et al., 2018; Mishra et al., 2020; Lou et al., 2021; Huang et al., 2022; Diaa et al., 2024). It keeps computational costs relatively low but requires multiple communication rounds, leading to higher communication and bandwidth requirements that limit applicability. Non-interactive PPML, typically using FHE, adopts a fire-and-forget paradigm (Gilad-Bachrach et al., 2016; Brutzkus et al., 2019; Lou & Jiang, 2021; Lee et al., 2022; Sarkar et al., 2023; Ao & Boddeti, 2024), where the client encrypts inputs, the provider computes on ciphertexts, and the client decrypts the output. This reduces communication to a single round at the expense of heavy server-side computation. Despite the costs, non-interactive PPML is well-suited to third-party services as it eliminates client involvement during inference, supports low-resource devices, and works in low-bandwidth settings. Motivated by these advantages, this work focuses on non-interactive PPML.

A key challenge in non-interactive PPML is implementing activation functions with cryptographic primitives. Activations are non-polynomial, while FHE supports only polynomial operations like addition and multiplication. Prior work addresses this by approximating activations with single polynomials (Diaa et al., 2024) or piecewise polynomials (Liu et al., 2017). These approximations can be introduced before or after training. In *post-training approximation*, models are trained with standard activations (e.g., ReLU) and later replaced with high-precision polynomials. For example, MPCNN (Lee et al., 2022) uses minimax polynomials of orders $\{15, 15, 27\}$. In *pre-training approximation*, ReLU is replaced with a polynomial before training, allowing lower-degree approximations that reduce multiplicative depth and speed up inference. For example, shallow CNNs like CryptoNets (Gilad-Bachrach et al., 2016) and LoLa (Brutzkus et al., 2019), similar to LeNet-5 (LeCun et al., 1998), achieve over 98% accuracy on MNIST with degree-2 polynomials. However, Garimella et al. (2021) showed that training larger models (beyond AlexNet (Krizhevsky et al., 2012) or VGG-11 (Simonyan & Zisserman, 2015)) with such low-degree polynomials causes large approximation errors that destabilize training. *To date, PILLAR (Diaa et al., 2024), which uses a degree-4 polynomial, is the lowest-degree approximation that generalizes to deep neural networks.*

Implementing non-interactive PPML becomes increasingly difficult as model depth increases. While *bootstrapping* enables evaluation of arbitrarily deep models by refreshing ciphertexts, it is an expensive operation in FHE and remains the main scalability bottleneck. To avoid this cost, prior work has relied on *Leveled Fully Homomorphic Encryption* (LFHE) (see §A for details). LFHE eliminates bootstrapping and allows faster private inference, but its computation is bounded by a fixed multiplicative depth determined by encryption parameters. As a result, the size and complexity of neural networks that can be evaluated under LFHE are strictly limited. *The deepest CNNs shown to run with LFHE alone are AlexNet and VGG-11*, as demonstrated by Garimella et al. (2021). Larger models such as ResNet-20 or ResNet-32 have so far required bootstrapping. For example, AutoFHE executes ResNet-20 inference with at least five bootstrapping layers (Ao & Boddeti, 2024), which leads to heavy computational costs. In this work, we focus on ResNet (He et al., 2016), as it is widely adopted in state-of-the-art evaluations and is well-suited for efficient and low-latency deployment.

**Contributions**: This work advances non-interactive PPML by enabling, for the first time, large CNNs like ResNet-18, ResNet-20, and ResNet-32 to run entirely under LFHE without bootstrapping, achieving accuracy comparable to plaintext ReLU models, while outperforming prior work in both accuracy and inference time. Our contributions are threefold:

- To the best of our knowledge, this is the first work to train large CNNs, like ResNet-18, ResNet-20, and ResNet-32 effectively with *degree-2 polynomial activations*. We introduce a novel *penalty function* that ensures stable training and accuracy on par with ReLU-based models. This achieves the theoretical minimum multiplicative depth of one for non-linear activations, whereas the lowest previously known stable alternative, PILLAR, required depth three (see §2).

- We introduce three novel *structural optimization* techniques that reduce the overall multiplicative depth of a model while preserving functional equivalence: (i) *Node Fusing*, which merges consecutive operations such as convolution and batch normalization by folding normalization parameters into convolution weights, eliminating separate multiplications; (ii) *Weight Redistribution*, which adjusts parameters so that coefficients of highest-order terms in polynomial activations and batch normalization, as well as divisors in pooling layers, normalize to one, removing redundant multiplications by constants; and (iii) *Tower Reuse*, which allows multiple multiplications within the same FHE level before rescaling. In the CKKS scheme used in this work (Cheon et al., 2017), ciphertexts carry a scale factor that grows with multiplications. Rescaling keeps ciphertext values within range but consumes a level. Since each level requires a modulus and homomorphic operations apply to all moduli, fewer rescaling steps reduce the number of moduli, leading to faster homomorphic operations. Moreover, they permit using larger moduli for higher accuracy (see §3).

- We propose an encoding strategy that utilizes unused ciphertext slots in FHE to pack multiple model instances, enabling simultaneous *ensemble inference* to improve accuracy at some additional inference cost. To offset this overhead, we introduce a *parameter clustering* method for convolutions that recovers much of the lost speed. To the best of our knowledge, this is the first work to exploit unallocated ciphertext slots for ensemble inference in FHE (see §4).

We evaluate our method on CIFAR-10 and CIFAR-100 datasets (Krizhevsky & Hinton, 2009), and show that it achieves up to $4\times$ faster private inference than prior work while achieving accuracy comparable to plaintext ReLU models (see §5). We will *open-source* our implementation.

## 2 MODEL TRAINING WITH POLYNOMIAL APPROXIMATION

### 2.1 PROBLEM SETTING

**Dataset.** We consider a dataset $\mathcal{D} = \{(x_i, y_i)\}_{i=1}^M$ for multi-class classification, where $x_i \in \mathbb{R}^n$ is the feature vector and $y_i \in \{1, \ldots, K\}$ is the class label. $\mathcal{D}$ has $M$ samples and $K$ classes.

**ReLU Network.** Let $f : \mathbb{R}^n \to \mathbb{R}^K$ be an $L$-layer feed-forward network with ReLU activations, parameterized by weight matrices $W^{(1)}, \ldots, W^{(L)}$, where each $W^{(l)} \in \mathbb{R}^{n_l \times n_{l-1}}$ and $n_l$ is the number of neurons in layer $l$. The network output is given by $f(x) = h^{(L)}(x)$, where $h^{(0)}(x) = x$ and $h^{(l)}(x) = ReLU(W^{(l)} h^{(l-1)}(x))$ for all $1 \leq l \leq L$. The activation function $ReLU(z)$ is applied coordinate-wise and defined as $ReLU(z) = \max(0, z)$.

**Polynomial Network.** We define $g : \mathbb{R}^n \to \mathbb{R}^K$ with the same architecture as $f$, but replace ReLU with a degree-$d$ polynomial $p_d(z) = \sum_{k=0}^d a_k z^k$ that approximates ReLU over a bounded interval $[-c, c]$. The polynomial coefficients $\{a_k\}_{k=0}^d$ are chosen so that the maximum approximation error satisfies: $\max_{z \in [-c,c]} |ReLU(z) - p_d(z)| \leq \varepsilon$, where $\varepsilon > 0$ is a small constant. The network output is given by $g(x) = h_{p_d}^{(L)}(x)$, where $h_{p_d}^{(0)}(x) = x$ and $h_{p_d}^{(l)}(x) = p_d(W^{(l)} h_{p_d}^{(l-1)}(x))$ for all $1 \leq l \leq L$.

**Challenges.** Using polynomial networks ensures FHE compatibility but introduces training challenges such as *escaping activations* and *coefficient truncation* (see §B for more details). Escaping activations occur when intermediate outputs drift outside the interval $[-c, c]$, leading to large approximation errors. Coefficient truncation occurs when polynomial coefficients are stored with limited fixed-point precision $b$, which changes the polynomial's shape and causes deviation from ReLU even inside the approximation interval. These issues require careful selection of $d$, $[-c, c]$, and $b$, together with regularization during training to keep activations within the valid approximation range.

### 2.2 TRAINING STRATEGY

**Quantization-Aware Polynomial Fitting.** To avoid coefficient truncation, we integrate fixed-point constraints directly into coefficient estimation using *quantization-aware polynomial fitting* (Diaa et al., 2024). Given an approximation interval $[-c, c]$ and fixed-point precision with $b$ fractional bits, we define a quantized input domain for polynomial fitting as: $X = \{x \in [-c, c] \mid x = k \cdot 2^{-b}, \ k \in \mathbb{Z}\}$. For each input $x_i \in X$, we compute scaled ReLU outputs $Y_i = 2^b \cdot ReLU(x_i)$ so that regression targets are integers, which reduces risk of precision loss during optimization. We build a matrix $B \in \mathbb{R}^{M \times (d+1)}$ with entries $B_{i,k} = x_i^k$ for $0 \leq k \leq d$. The coefficients $A = [a_0, \ldots, a_d]^\top \in \mathbb{Z}^{d+1}$ are obtained by solving bounded integer least-squares problem: $\min_{A \in \mathbb{Z}^{d+1}} \|BA - Y\|_2^2$ subject to $a_k \in [-2^{b-1}, 2^{b-1} - 1]$. The resulting fixed-point polynomial is $p_d(x) = \sum_{k=0}^d (\frac{a_k}{2^b}) x^k$.

**Activation Regularization.** To avoid escaping activations, we introduce a *regularization strategy* that constrains the inputs to polynomial activation functions (i.e., the pre-activations) to remain within the valid approximation interval $[-c, c]$. The classification loss supervises only the final output and provides no mechanism to limit intermediate pre-activation values. Hence, minimizing it alone does not prevent pre-activations from drifting outside the interval $[-c, c]$, where the polynomial diverges from ReLU and destabilizes training. We introduce a layer-wise penalty that penalizes out-of-range pre-activations. For a mini-batch $\mathcal{B} \subset \mathcal{D}$, the following training loss is minimized:

$$\mathcal{L}_\mathcal{B} = \underbrace{\frac{1}{|\mathcal{B}|} \sum_{(x,y) \in \mathcal{B}} \ell_{CE}(g(x), y)}_{\text{classification loss}} + \zeta \underbrace{\frac{1}{L} \sum_{l=1}^L \left\| z_p^{(l)} - \text{clip}\left(z_p^{(l)}; [-c, c]\right) \right\|_2}_{\text{clip-range penalty}} \qquad (1)$$

where $\ell_{CE}$ is the *cross-entropy loss*, $z_p^{(l)}$ are pre-activations at layer $l$, i.e., outputs of affine transformations $W^{(l)} h_{p_d}^{(l-1)}(x)$ before the polynomial activation. The function $\text{clip}(z; [-c, c])$ is applied element-wise and defined as: $\text{clip}(z; [-c, c])_i = \max(-c, \min(z_i, c))$. The second loss term penalizes pre-activations that lie outside the interval $[-c, c]$, with strength controlled by the regularization parameter $\zeta > 0$. See §C for a proof of the penalty function's correctness. Training can still become unstable for two reasons: (i) in early epochs pre-activations can grow unbounded before the model learns to contain them and (ii) using full regularization weight $\zeta$ from the start can let the penalty dominate and destabilize optimization. As additional strategies, we also consider *clipping pre-activations during training* and introducing a *warm-up schedule for $\zeta$* (see §D for more details).

## 3 STRUCTURAL OPTIMIZATIONS

### 3.1 NODE FUSING

Our first optimization is *node fusing*, which merges consecutive computational nodes into a single equivalent function. This reduces both number of operations and multiplicative depth. Formally, two sequential nodes $F(G(x))$ are replaced with $H(x)$ such that $F(G(x)) = H(x)$, with $H(x)$ more efficient to evaluate. Since our method uses batch normalization, recall its polynomial form $B(x) = b_1 x + b_0$, $b_1 = \frac{\gamma}{\sigma}$, $b_0 = \beta_b - b_1 \mu$, where $\mu, \sigma$ are the input mean and standard deviation, and $\gamma, \beta_b$ are learnable parameters. We outline four common fusion cases. See §E.1 for full derivations.

**Case 1:** $P(B(x)) \mapsto P(x)$**.** When a polynomial activation $P(\cdot)$ follows batch normalization, the two can be fused into a single quadratic polynomial. If $P(x) = c_2 x^2 + c_1 x + c_0$, then $P(B(x)) = p_2 x^2 + p_1 x + p_0$, with fused coefficients $p_2 = b_1^2 c_2$, $p_1 = b_1(2b_0 c_2 + c_1)$, $p_0 = b_0^2 c_2 + b_0 c_1 + c_0$.

**Case 2:** $B(C(x)) \mapsto C(x)$**.** When batch normalization follows a convolution $C(\cdot)$, the two can be fused (Markuš, 2018). A convolution is given by $C(x) = \sum_i w_i x_i + \beta_c$, where $w_i$ are the convolution weights and $\beta_c$ is the bias. The fused form becomes a single convolution $B(C(x)) = \sum_i \omega_i x_i + \alpha$, with fused coefficients $\omega_i = b_1 w_i$, $\alpha = b_1 \beta_c + b_0$.

**Case 3:** $P(B_X(x) + B_Y(y)) \mapsto S(x, y)$**.** In residual networks, skip connections often merge two batch-normalized branches by summing them before applying an activation $P(\cdot)$. Specifically, $P(B_X(x) + B_Y(y))$. Here, $B_X$ and $B_Y$ are independent batch normalization layers to input $x$ and $y$. This structure can be fused into a single quadratic bivariate polynomial, which we call *polyskip*: $S(x, y) = d_{X2} x^2 + d_{Y2} y^2 + d_{XY} xy + d_X x + d_Y y + d_0$, with coefficients $d_{X2} = c_2 b_{X1}^2$, $d_{Y2} = c_2 b_{Y1}^2$, $d_{XY} = 2c_2 b_{X1} b_{Y1}$, $d_X = b_{X1}(2c_2(b_{X0} + b_{Y0}) + c_1)$, $d_Y = b_{Y1}(2c_2(b_{X0} + b_{Y0}) + c_1)$, $d_0 = c_2(b_{X0} + b_{Y0})^2 + c_1(b_{X0} + b_{Y0}) + c_0$.

**Case 4:** $P(B_X(x) + y) \mapsto S(x, y)$**.** Some skip connections use identity shortcuts, where the input $y$ is added directly to the batch-normalized branch $B_X(x)$ before applying the activation. This structure can be fused into a quadratic bivariate polynomial: $S(x, y) = d_{X2} x^2 + d_{Y2} y^2 + d_{XY} xy + d_X x + d_Y y + d_0$,with coefficients $d_{X2} = c_2 b_{X1}^2$, $d_{Y2} = c_2$, $d_{XY} = 2c_2 b_{X1}$, $d_X = b_{X1}(2c_2 b_{X0} + c_1)$, $d_Y = 2c_2 b_{X0} + c_1$, $d_0 = c_2 b_{X0}^2 + c_1 b_{X0} + c_0$.

Node fusing collapses batch normalization, activations, convolutions, and skip connections into single fused polynomials, eliminating redundant nodes and reducing multiplicative depth.

### 3.2 WEIGHT REDISTRIBUTION

Our second optimization is *weight redistribution*, a technique that redistributes weights across the network while maintaining functional equivalence. The goal is to reduce the multiplicative depth of certain functions by one. It specifically targets average pooling, batch normalization, and polynomial activations. For average pooling, redistribution eliminates the normalization step by setting the divisor to one. For polynomial functions, the highest-order coefficient is set to one. For example, the polynomial activation $c_2 x^2 + c_1 x + c_0$ is transformed into $x^2 + p_1 x + p_0$, reducing its multiplicative depth from two to one, *the theoretical minimum for an activation function*. These transformations alone break model equivalence. To maintain it, other nodes in the network must compensate for the change. We refer to the nodes initiating weight redistribution as *donors*, and the nodes adjusting their parameters to preserve equivalence as *receivers*. We represent the network as a directed graph and traverse it, identifying all eligible donor nodes. For each donor, redistribution can be applied either *forward* or *backward*, affecting receivers among the donor's successors or predecessors, respectively.

**Update Forward.** *Donors.* We first consider donor updates in the forward direction. Our goal is to construct a normalized function $\bar{F}(x)$ such that $\upsilon \bar{F}(x) = F(x)$, where $F(x)$ is the original donor function and $\upsilon$ is the update term to be propagated to receivers. The average pooling operation is given by $\mu(x) = k^{-1} \sum_i x_i$, where $k$ is the kernel size. In the normalized function we set $k = 1$, reducing the operation to $\bar{\mu}(x) = \sum_i x_i$. Thus, for the equality to be valid, we have $\upsilon = k^{-1}$.

Batch normalization and polynomial activation are polynomial functions. For a univariate polynomial $P(x) = \sum_{i=0}^{d} c_i x^i$ of degree $d$, and a normalized polynomial $\bar{P}(x) = x^d + \sum_{i=0}^{d-1} \bar{c}_i x^i$, where $\bar{c}_i$ are the normalized coefficients, we have $\upsilon = c_d$ and $\bar{c}_i = c_i \upsilon^{-1}$ for $\upsilon \bar{P}(x) = P(x)$ to hold. The extension to the bivariate polynomial activation is analogous and deferred to §E.2.

*Receivers.* After a donor update, receivers must be adjusted to maintain model equivalence. The original composed function is $G(F(\cdot))$, where $F(\cdot)$ is the donor and $G(\cdot)$ is the receiver. Updating the donor to $\bar{F}(\cdot)$ without compensating the receiver results in $G(v^{-1}F(\cdot))$, breaking model equivalence. To restore it, we modify the receiver so that $\bar{G}(\bar{F}(\cdot)) = \bar{G}(v^{-1}F(\cdot)) = G(F(\cdot))$.

Receivers fall into two categories: kernel functions and polynomial functions. Kernel functions share the form $K(x) = \sum_i w_i x_i + \beta$. To maintain model equivalence, we must update the kernel function such that $\bar{K}(v^{-1}x) = K(x)$. For that, we must set $\bar{w}_i = w_i v$ and $\bar{\beta} = \beta$.

Polynomial functions update as $\bar{P}(v^{-1}x) = P(x)$, which implies that $\bar{c}_i = c_i v^i$. For the bivariate case, in $\bar{S}(v^{-1}x, y) = S(x, y)$ the index $i$ corresponds to the exponent of $x$, while in $\bar{S}(x, v^{-1}y) = S(x, y)$ it corresponds to the exponent of $y$.

**Update Backward.** *Donors.* Backward updates are essentially the inverse of forward updates. For average pooling, the update is identical, since $\bar{\mu}(vx) = v\bar{\mu}(x)$, while for polynomial donors, it must hold that $\bar{P}(vx) = P(x)$; hence, $v = c_d^{1/d}$ and $\bar{c}_i = c_i v^{-i}$.

*Receivers.* Receivers must satisfy $\bar{G}(x) = vG(x)$ to preserve model equivalence. Kernel receivers are updated as $\bar{K}(x) = vK(x)$, which implies that $\bar{w}_i = w_i v$ and $\bar{\beta} = \beta v$. And polynomial receivers update according to $\bar{P}(x) = vP(x)$; therefore, $\bar{c}_i = c_i v$.

## 3.3 Tower Reuse

In CKKS, the Residue Number System (RNS) moduli are typically chosen as primes close to the scaling factor $\Delta$ (see §A for an overview of LFHE). This choice is motivated by the fact that, after each homomorphic multiplication, the scaling factors of the operands are multiplied, producing a ciphertext with scale $\Delta^2$. To control this growth, the *rescale* operation reduces the scale back to $\Delta$ by removing one modulus from the modulus chain. However, rescaling introduces approximation errors: the further the dropped modulus deviates from $\Delta$, the larger the error. These errors may accumulate and amplify through subsequent homomorphic operations. A larger $\Delta$ is desirable for two reasons: (i) it provides higher precision for CKKS encoding and (ii) the RNS moduli are relatively closer to the scaling factor, reducing approximation errors.

The drawback is that a larger $\Delta$ requires larger RNS moduli, which in turn increases the ciphertext modulus $Q = \prod_{i=0}^{\mathcal{L}} q_i$, where $\mathcal{L}$ is the number of levels and $q_i$ are the individual RNS primes. Since the security level depends on the ratio $N/Q$ (with $N$ the polynomial degree), a larger $Q$ reduces security. One way to increase security is to increase $N$, but this slows computations due to larger polynomials. Thus, for a fixed security level and polynomial degree, there is a maximum allowable $Q$. Consequently, for an application-defined multiplicative depth $\delta$, this imposes a bound on the maximum scaling factor $\Delta$. In practice, when $\delta$ is large, as is common in deep learning models, the resulting $\Delta$ is small. This has two adverse effects, opposite to the benefits of a large $\Delta$: lower numerical precision and larger approximation errors, as moduli $q_i$ are relatively farther from $\Delta$.

**Proposed Solution.** We introduce a more general method for determining the RNS moduli by introducing *sublevels* within each level. Instead of enforcing $q_i \approx \Delta$, we allow $q_i \approx \Delta^\ell$, where $\ell$ denotes the number of sublevels. Rescaling is performed only when a ciphertext scale exceeds the sublevel capacity of the modulus to be dropped. Formally, $\lambda(x) > \lambda(q_i) \implies \downarrow x$, where $\downarrow$ denotes the rescale operation, and $\lambda(x) = \lfloor \log_\Delta \Delta_x \rceil$ returns the sublevel of a ciphertext, plaintext, or modulus: with $\Delta_x$ referring to the scale of $x$ and $\Delta$ being the default scale defined by the encryption parameters. See §E.3 for an example of this technique.

This method allows smaller $\Delta$ values while maintaining larger $q_i$, mitigating rescaling errors. Moreover, fewer effective levels (moduli) reduce error amplification, and since homomorphic operations are applied across all moduli, reducing their number improves computational efficiency.

## 3.4 Impact on RNS Levels

The combination of our polynomial activation function with the structural optimizations proposed in this section reduces the number of RNS levels $\mathcal{L}$ required for multiplication compared to PILLAR (Diaa et al., 2024), which previously achieved the lowest $\mathcal{L}$: from 87 to 18 for ResNet-18, from 97 to 20 for ResNet-20, and from 157 to 32 for ResNet-32. A complete analysis of the contribution of each technique to the reduction in required levels for ResNet models is provided in §E.4.

## 4 CODESIGN TECHNIQUES

### 4.1 DATA LAYOUT

*Data layout* defines how model inputs and weights are mapped into FHE ciphertext and plaintext slots during encoding. In this work, we adopt the *HW layout* (Dathathri et al., 2019), which assigns each input channel to a separate ciphertext and fills its slots with spatial values indexed by two–dimensional coordinates. Unlike the naive layout (Gilad-Bachrach et al., 2016), where each ciphertext holds only one value and incurs significant overhead, HW layout is more efficient, packing $h \cdot w$ values per ciphertext, with $h$ and $w$ denoting input height and width. Figure 1 shows this for a $3 \times 3 \times 3$ input with two $3 \times 2 \times 2$ filters (stride 1, no padding), illustrating slot arrangements and gaps that may appear after convolution or other kernel operations. These gaps could be removed by remapping, but that requires additional multiplications and increases multiplicative depth. Instead, we use an *adaptive (lazy) mapping* strategy: some slots are left unused, and subsequent weights, biases, and coefficients are aligned with the slot arrangement of the preceding layer's output. HW layout does not always fully utilize slots, especially when input channels have far fewer values than available slots. More compact layouts such as *CHW layout* (Dathathri et al., 2019) and its variants (Lee et al., 2022) address this by mapping multiple channels into one ciphertext, which reduces ciphertext counts and decreases the number of multiplications and additions, ultimately leading to faster inference. However, PPML relies on polynomial approximations that reduce accuracy compared to plaintext ReLU models. For this reason, we retain the less efficient HW layout, repurposing unused slots to improve accuracy instead of minimizing inference time. To offset the added latency, we introduce a parameter clustering technique.

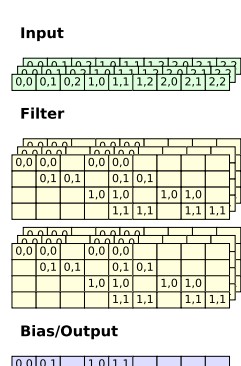

Figure 1: Illustration of HW layout for convolution on a $3 \times 3 \times 3$ input and two $3 \times 2 \times 2$ filters, configured with padding 0 and stride 1.

### 4.2 CLUSTERING OF CONVOLUTION PARAMETERS

Convolution layers account for most parameters in deep neural networks, and their weight handling under FHE is costly since each weight must be repeatedly encoded to match varying ciphertext layouts and levels. As these depend on kernel position and layer structure, the same scalar weight is often redundantly encoded. Pre-encoding all weight–layout combinations would lead to prohibitive memory usage, while on-demand encoding reduces memory but adds heavy computational overhead. To mitigate this, weights are limited to a small fixed set of representative values. Each representative is encoded once into a plaintext codebook, and during inference, the appropriate plaintext is retrieved rather than recomputed. Time and memory then scale with codebook size instead of the number of weights. Clustering provides a way to build this representative set by grouping similar weights and replacing each with its closest *centroid*, bounding the number of unique encodings.

Formally, let a polynomial network contain $L$ convolution layers. The weight tensor of layer $l$ is $W^{(l)} \in \mathbb{R}^{O_l \times I_l \times H_l \times W_l}$, where $O_l$ is the number of output channels, $I_l$ the number of input channels, and $H_l \times W_l$ the kernel size. Flattening all convolution weights gives $\theta = (\theta_1, \ldots, \theta_P) \in \mathbb{R}^P$, where $P$ is the total number of parameters. To compress the model, each $\theta_j$ is replaced by its nearest codebook value from $\mathcal{C} = \{c_1, \ldots, c_k\} \subset \mathbb{R}$, where $k \ll P$ is the number of representatives in the codebook. The codebook entries act as *centroids* approximating the original weights. The choice of $k$ determines the efficiency-accuracy tradeoff: smaller $k$ reduces the number of encodings but increases approximation error, while larger $k$ produces finer approximation at higher computational and memory cost. Given a distance function $d : \mathbb{R} \times \mathbb{R} \to \mathbb{R}$ (assumed to be non-negative), each parameter is quantized as: $\tilde{\theta}_j = \arg\min_{c \in \mathcal{C}} d(\theta_j, c)$ for all $j = 1, \ldots, P$, producing a quantized parameter vector $\tilde{\theta} \in \mathcal{C}^P$. Quantization is applied elementwise, so each convolutional kernel retains its shape while its values are drawn from the codebook. The choice of clustering strategy determines how the codebook is constructed and how assignments are made.

**Full Clustering.** A straightforward strategy is to cluster the entire parameter vector $\theta \in \mathbb{R}^P$ using a single global codebook $\mathcal{C}$. All convolution parameters, regardless of layer or kernel position, are quantized to the same shared set of representative values. This produces a uniform quantization scheme with easy implementation and a small codebook size.

*Limitation.* While full clustering reduces the number of distinct weight values, it does not eliminate repeated plaintext encodings. Each weight must still match the ciphertext layout and level of its layer, so the same scalar value appearing in different layers must be re-encoded. Even within a single convolution, all weights share the same level, but weights from different filter columns map to different layouts and require separate encodings. Only weights in the same column across layer filters align in layout and level. We define this group as a *slice*, which can share plaintext encoding. Thus, full clustering reduces codebook size but not redundancy from mismatched levels and layouts.

**Slice Clustering.** To address the redundancy remaining under full clustering, we refine quantization to the slice level. Since only weights within the same slice can share plaintext encodings, clustering is applied per slice rather than across all parameters. For a given layer $l$ and spatial position $s \in \{1, \ldots, W_l\}$, the slice is $S_s^{(l)} = \{W_{o,i,h,s}^{(l)} | o = 1, \ldots, O_l; \ i = 1, \ldots, I_l; \ h = 1, \ldots, H_l\}$, which can be viewed as a vector in $\mathbb{R}^{O_l I_l H_l}$ (see §F.1 for an illustration). Each slice $S_s^{(l)}$ is clustered independently with its own codebook $\mathcal{C}_s^{(l)} \subset \mathbb{R}$, and quantization is applied elementwise: $\tilde{W}_{o,i,h,s}^{(l)} = \arg \min_{c \in \mathcal{C}_s^{(l)}} d(W_{o,i,h,s}^{(l)}, c)$. This ensures clustering respects FHE structure while adapting to the local weight distribution of each slice. The additional storage from slice-specific codebooks is modest compared to the resulting gains in accuracy and efficiency.

## 4.3 ENSEMBLE OF POLYNOMIAL NETWORK

Inference with a single polynomial network $g(x)$ often shows high variance across training runs. Let $g_{(m)}(x) \in \mathbb{R}^K$ be the logits from the $m$-th independently trained instance. For a fixed input, these logits can differ significantly, especially near decision boundaries, due to (1) approximation error from the fixed-point polynomial activation $p_d(\cdot)$ and (2) training stochasticity such as random weight initialization and mini-batch ordering. As a result, different $g_{(m)}$ may predict different classes for the same input. To reduce this variance, we use an ensemble of $M$ polynomial networks $\{g_{(m)}\}_{m=1}^M$. Each has the same architecture and activation $p_d(\cdot)$ but is trained independently with different seeds and mini-batch orders. For input $x$, the ensemble output is the average of logits: $\overline{g}(x) = \frac{1}{M} \sum_{m=1}^M g_{(m)}(x)$ with the predicted class $\hat{y} = \arg \max_{k \in \{1, \ldots, K\}} \overline{g}(x)_k$. Averaging smooths training noise and polynomial approximation errors, producing more stable and accurate predictions. Each $g_{(m)}$ is trained with the regularized loss $\mathcal{L}_\mathcal{B}$ (Eq. 1), ensuring compatibility with fixed-point polynomial inference under FHE. Crucially, this ensemble design adds no extra computation or memory overhead. As noted in §4.1, our HW layout leaves some ciphertext slots unused. We fill these slots with weights from different models, while all ensemble members share the same ciphertexts for inputs and activations. This reuse of ciphertext–plaintext structures avoids redundant ciphertexts and repeated encodings, making ensemble inference practical in the FHE framework.

**Limits of Ensemble with Clustering.** While ensemble inference reuses unused ciphertext slots, its combination with parameter clustering creates a challenge. In slice clustering, each slice $S_s^{(l)}$ with $O_l \times I_l \times H_l$ weights is represented by $k$ centroids in codebook $\mathcal{C}_s^{(l)}$, reducing plaintext encodings to $k$ per slice. In an ensemble, however, each plaintext must encode parameters from all $M$ models. As centroids are unlikely to align across models, the count of unique encodings per slice grows as $\mathcal{O}(k^M)$, canceling the gains of clustering and making the naive combination impractical.

**Slice Ensemble Clustering.** To address the inefficiency of independent clustering, we extend slice clustering to ensembles by enforcing shared representatives across models. We cluster weights jointly at the same kernel position so that all models use a common codebook. Formally, the ensemble has $M$ polynomial networks. For convolution layer $l$, model $m$ has weights $W^{(l,m)} \in \mathbb{R}^{O_l \times I_l \times H_l \times W_l}$. For kernel position $s \in \{1, \ldots, W_l\}$, we extract slices: $S_s^{(l,m)} = \{W_{o,i,h,s}^{(l,m)} | o = 1, \ldots, O_l; i = 1, \ldots, I_l; h = 1, \ldots, H_l\}$. Stacking slices across $M$ models produces $X_s^{(l)} = [S_s^{(l,1)} \ S_s^{(l,2)} \cdots S_s^{(l,M)}] \in \mathbb{R}^{N_s \times M}$, where $N_s = O_l I_l H_l$. Each row is an $M$-dimensional vector for the same weight coordinate across models (see §F.2 for an illustration). We cluster rows in $\mathbb{R}^M$: $\min_{\mathcal{C}_s^{(l)} \subset \mathbb{R}^M} \sum_{j=1}^{N_s} \min_{c \in \mathcal{C}_s^{(l)}} d(X_{s,j}^{(l)}, c)$, with codebook $\mathcal{C}_s^{(l)} = \{c_1, \ldots, c_k\}$. Each $X_{s,j}^{(l)}$ is replaced by its nearest centroid, producing quantized slices $\tilde{X}_{s,j}^{(l)} \in \mathcal{C}_s^{(l)}$. Quantized weights $\tilde{W}^{(l,m)}$ are reconstructed by reshaping slices. This shared clustering aligns weights across models, mapping small variations at the same coordinate to a common centroid. It thus requires fewer distinct encodings, reducing codebook size and inference time while retaining the accuracy gains of ensemble.

## 5 EXPERIMENTAL RESULTS

### 5.1 EXPERIMENTAL SETUP

We trained ResNet-18, ResNet-20, and ResNet-32 on CIFAR-10 and CIFAR-100 using PyTorch 2.4.1. Each experiment was repeated 10 times, and we report mean accuracy and private inference time with standard deviations. Private inference used a C++20 framework we developed, built on Microsoft SEAL 4.1 (SEAL) for leveled FHE operations and GMP 6.2.1 for multiprecision arithmetic. The framework includes a tool that automatically converts trained models for C++ inference. We use a machine with two AMD EPYC 64-core processors, 2 TB memory across 32 DDR4 DIMMs at 2933 MT/s, on Ubuntu 22.04.5 LTS with kernel 5.15.0-151-generic (x86_64) to run all experiments, including related work. For model training (see §2), we used hyperparameters $c = 2$ and $\zeta = 0.001$. An ablation study for both is provided in §G. We set $b = 10$, aligned with PILLAR (Diaa et al., 2024), as it provides accurate polynomial approximation after quantization-aware fitting. For all clustering strategies, we used the $\ell_2$ norm as the distance metric, with *k-means* as the clustering algorithm (Lloyd, 1982). Details on FHE encryption parameters are in §H.1. Additional results, including peak memory, are reported in §H.3.

We compare our approach against non-interactive PPML baselines, MPCNN (Lee et al., 2022) and AutoFHE (Ao & Boddeti, 2024), in terms of accuracy and inference time. For fairness, we use their publicly available implementations (git, 2024a; 2022) with specified encryption parameters. Both baselines implement ResNet-20 on CIFAR-10 and ResNet-32 on CIFAR-10 and CIFAR-100, all with $N = 2^{16}$. We also trained plaintext models using standard ReLU activations to serve as a reference for accuracy. In this section, we mainly report results for ResNet-20 and ResNet-32, since these allow direct comparison with baselines. Results for ResNet-18 are provided in §H.2.

### 5.2 SUMMARY OF RESULTS

**Accuracy of Polynomial Approximation.** Training with the polynomial approximation described in §2 shows that, on average across CIFAR-10 and CIFAR-100, *our degree-2 polynomial achieves accuracy only about 1.3% lower than ReLU*, while PILLAR with degree-4 polynomial is about 5.4% lower. A degree-2 adaptation of PILLAR shows an average drop of 6.5%. These results demonstrate that *our method can use low-degree polynomials effectively*, while PILLAR does not reach comparable accuracy even with higher degrees. A detailed comparison is provided in §G.3.

**Analysis of Parameter Clustering.** Figure 2 (top row) shows accuracy and inference time for three methods: *Standard*, *Full Clustering*, and *Slice Clustering*. *Standard* applies techniques in §2 and §3, while *Full Clustering* and *Slice Clustering* extend it with parameter clustering as described in §4.2. Clustering reduces inference time several-fold compared to *Standard*, while maintaining similar accuracy. At very low centroid counts ($k$), accuracy reduces due to insufficient parameter representation. Beyond a threshold (e.g., $k = 64$ for *Slice Clustering*), accuracy matches *Standard* while providing 4-7× speedups. For fixed $k$, *Full Clustering* is slightly faster since each layer does not utilize all centroids, whereas *Slice Clustering* uses all centroids within each slice, introducing additional computational overhead from extra plaintext encoding. However, *Slice Clustering* provides finer granularity and better parameter representation, leading to higher accuracy. In fact, *Slice Clustering with $k = 32$ achieves higher accuracy and faster inference than Full Clustering with $k = 64$*. Overall, *Slice Clustering* is superior, surpassing *Full Clustering* in both accuracy and latency.

**Ensemble Evaluation.** We evaluate ensembles with $M = \{1, 2, 4\}$ models. Figure 2 (middle row) shows accuracy and inference time for two approaches: *Standard-M* (ensembles without clustering) and *Slice-M* (ensembles with slice clustering as in §4.3). Inference time does not increase with ensemble size since all models fit in one ciphertext, requiring the same number of homomorphic operations. For *Standard-M*, accuracy consistently improves as $M$ increases. *Slice-M*, however, does not always scale the same way. Since each centroid represents an $M$-dimensional space, higher $M$ can require more centroids to capture model parameters. Thus, *Slice-4* can underperform *Slice-2* at a fixed $k$, and *Slice-2* needs more centroids to match *Standard-2* accuracy. Despite this, *Slice-M* provides notable advantages. *Slice-2* mostly outperforms *Standard-2* and reaches accuracy comparable to *Standard-4*. This suggests clustering acts as an implicit regularizer, mitigating overfitting.

**Comparison with Related Work.** Figure 2 (bottom row) presents Pareto fronts of *Slice Clustering* in terms of accuracy and private inference time, compared with AutoFHE and MPCNN. For reference, we also report *Standard-4* and plaintext ReLU accuracy. For ResNet-20, where our models operate with $N = 2^{15}$, *Slice Clustering* achieves consistently faster inference than related work (up

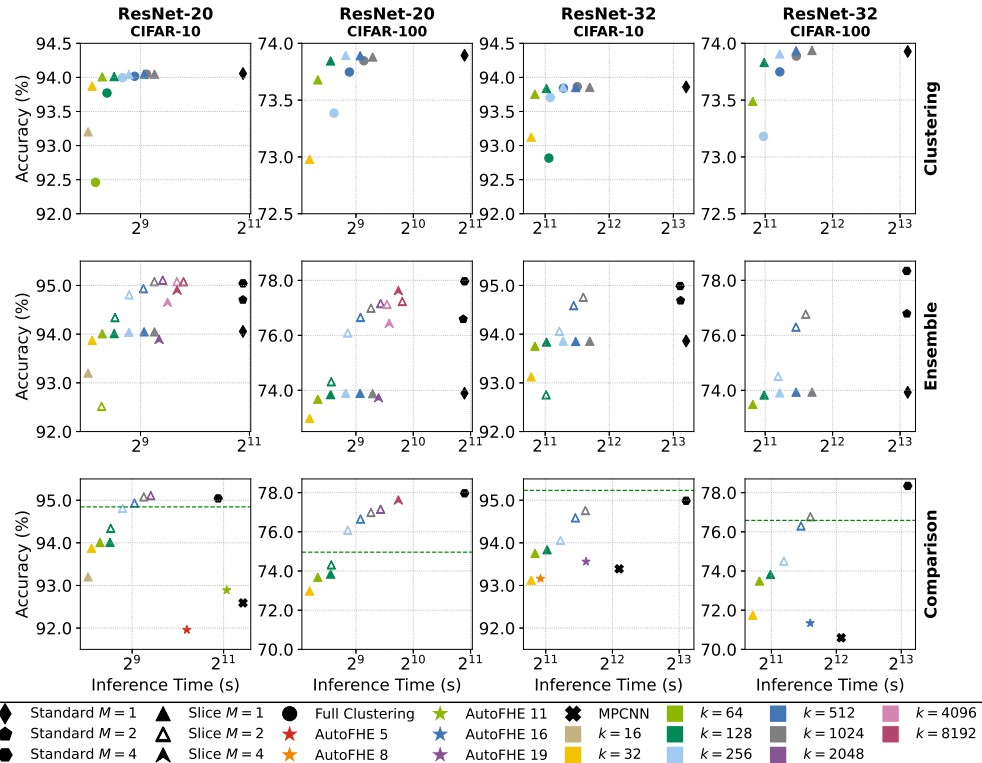

Figure 2: Accuracy vs. private inference time for ResNet-20/32 on CIFAR-10/100. Ensembles are evaluated with $M \in \{1, 2, 4\}$ ($M = 1$ is a single model). We compare Standard ensembles (no clustering), Slice ensembles (slice-wise clustering), and Full Clustering. Colored squares represent centroid counts ($k = 16, \ldots, 8192$), and the same color is used for the same centroid count in both slice and full clustering. Baselines include AutoFHE (numbers = bootstrapping layers) and MPCNN. The green dashed line shows plaintext ReLU accuracy.

to $4\times$). For ResNet-32, where all approaches use $N = 2^{16}$, our method is comparable in inference time to AutoFHE on CIFAR-10 and $2\times$ faster on CIFAR-100. Inference time does not change for our approach for different datasets, whereas AutoFHE slows down because it requires a high number of bootstrapping operations to maintain accuracy. The main factors affecting our inference time relative to related work are: The main factors affecting our inference time relative to related work are: (i) the polynomial activation (§2) and structural optimizations (§3) reduce both multiplicative depth and number of RNS moduli, allowing execution under LFHE without bootstrapping; (ii) our clustering technique (§4.2) further reduces the number of unique parameters, thereby limiting plaintext encodings per convolution; (iii) for ResNet-20 (and ResNet-18), we operate at a smaller polynomial degree, which directly accelerates homomorphic operations; (iv) our choice of data layout (§4.1) deliberately sacrifices some latency to better exploit ciphertext slots for accuracy, but above optimizations offset this overhead. In terms of accuracy, our approach consistently outperforms related work thanks to our training strategy (§2). In fact, it closely matches plaintext ReLU models, effectively eliminating the accuracy gap typically observed in polynomial approximations of ResNets.

## 6 CONCLUSION

This work significantly advances the practicality of PPML under FHE. We demonstrated that by integrating low-degree polynomial activations with structural and co-design optimizations, it is possible to execute large-scale models such as ResNet-18, ResNet-20, and ResNet-32 entirely under leveled FHE without bootstrapping, an achievement previously considered not possible. State-of-the-art methods typically incur a 2-5% accuracy loss compared to plaintext models. Our method is the first to close this gap, achieving accuracy on par with ReLU baselines while delivering up to $4\times$ faster private inference on CIFAR-10 and CIFAR-100. These results mark an important step toward making PPML deployable in real-world applications, particularly in domains such as healthcare and finance, where data confidentiality is non-negotiable. Future research will extend these methods to larger datasets and deeper architectures to further expand the practicality of PPML.

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

## THE USE OF LARGE LANGUAGE MODELS (LLMS)

We used LLMs only to improve the wording and readability of the paper. They were not involved in generating ideas, analysis, or writing beyond basic language edits.

## A  LEVELED FULLY HOMOMORPHIC ENCRYPTION

*Homomorphic Encryption* is a special type of encryption that allows meaningful computations to be performed directly on encrypted data. In this work, we employ the *Cheon-Kim-Kim-Song* (CKKS) (Cheon et al., 2017) encryption scheme, specifically the Residue Number System (RNS) variant (Cheon et al., 2018), which supports approximate arithmetic over encrypted real or complex numbers. CKKS is an FHE scheme based on the Ring-Learning With Errors (RLWE) problem (Lyubashevsky et al., 2013). The scheme operates across three domains: message $\mathcal{M} = \mathbb{C}$, plaintext $\mathcal{P} = R_Q = \mathbb{Z}_Q[x]/(x^N + 1)$, and ciphertext $\mathcal{C} = R_Q \times R_Q$ spaces. The encoding function $encode(\cdot) = \mathcal{M}^{N/2} \mapsto \mathcal{P}$ maps a vector of $N/2$ complex numbers into a degree-$N$ polynomial in the plaintext space. The encryption function $encrypt(\cdot) = \mathcal{P} \mapsto \mathcal{C}$ transforms a plaintext into a ciphertext. In RNS-CKKS, the ciphertext modulus $Q$ is decomposed into $\mathcal{L} + 1$ smaller primes $Q = \prod_{i=0}^{\mathcal{L}} q_i$, where $\mathcal{L}$ denotes the level. A polynomial in $R_Q$ is represented as $\mathcal{L} + 1$ polynomials in $R_{q_i} \forall i \in [0, \mathcal{L}] \cap \mathbb{Z}$, which allows computations using native 64-bit integer arithmetic rather than costly arbitrary-precision operations.

CKKS supports three homomorphic operations: addition, multiplication, and rotation. Additions and multiplications act element-wise on the encoded vector, similar to SIMD vector operations. The rotation operation rotates the vector of $N/2$ encoded messages by a step $s \in \mathbb{Z}_{N/2}$. During encoding, coefficients are scaled by a factor $\Delta$, rounded to the nearest integers, and reduced modulo $Q$, making CKKS a fixed-point FHE scheme. Multiplying two encoded messages scales the result by $\Delta^2$, requiring a *rescale* operation to restore the scale to $\Delta$. Rescaling removes one RNS modulus, reducing the level by 1. It also accumulates approximation errors, since the RNS modulus being discarded and the scaling factor $\Delta$ are not equal. When the level reaches zero, *bootstrapping* can reset the ciphertext to a higher level, enabling unbounded computation. However, bootstrapping is computationally expensive and avoided when possible. FHE without bootstrapping is called *Levelled Fully Homomorphic Encryption* (LFHE). In LFHE, the encryption parameters $(N, Q)$ are chosen to provide the desired security and sufficient levels for a predefined arithmetic circuit with known multiplicative depth. To the best of our knowledge, this work is the first to apply LFHE to deep CNN models like ResNet-20 and ResNet-32.

## B  CHALLENGES OF USING POLYNOMIAL ACTIVATION

ReLU, defined as $ReLU(x) = \max(0, x)$, is one of the most widely used activation functions in deep learning due to its simplicity and computational efficiency. Its piecewise linear structure introduces the non-linearity necessary for neural networks to capture complex patterns in data. While ReLU is straightforward to implement in plaintext settings, it poses significant challenges in FHE environments due to its reliance on conditional branching. FHE can evaluate only polynomial functions as it supports only additions and multiplications, thus, any FHE representation of ReLU must be a polynomial approximation. Evidently, the higher degree the polynomial, the more precise the approximation. However, the multiplicative depth of the activation functions grows logarithmically to the degree of the polynomial used for approximation. Precisely, the multiplicative depth can be computed as $\delta = \lceil \log_2 d + 1 \rceil$, where $d$ denotes the polynomial degree and the $+1$ refers to the coefficient multiplication. In the literature, the lowest multiplicative depth for a polynomial approximation of ReLU is achieved by PILLAR (Diaa et al., 2024), which uses a degree-4 polynomial of the form $\sum_{i=0}^{4} c_i x^i$ with $\delta = 3$. The theoretical minimum multiplicative depth for an activation function is one, achieved with a polynomial of the form $x^2 + c_1 x + c_0$. Polynomials like this have been effectively used in shallow CNNs like CryptoNets (Gilad-Bachrach et al., 2016) and LoLa (Brutzkus et al., 2019), but failed to work for deeper models due to the escaping activation problem (Garimella et al., 2021). In addition, polynomial approximations are inherently limited to bounded intervals of the input domain. When substituted directly for ReLU during model training, these approximations often lead to a significant drop in model accuracy (Garimella et al., 2021; Hussain et al., 2021; Diaa et al., 2024). This section analyzes two primary causes of this degradation: *escaping activations*,

where intermediate layer outputs fall outside the polynomial's approximation interval and *coefficient truncation*, which results from representing polynomial coefficients in fixed-point format to ensure compatibility with FHE arithmetic.

## B.1 Escaping Activation Problem

Let $p_d(x) = \sum_{k=0}^{d} a_k x^k$ denote a degree-$d$ polynomial approximation of the ReLU function, where each coefficient $a_k \in \mathbb{R}$ for all $k \in \{0, \ldots, d\}$. These coefficients are typically obtained by minimizing the least-squares error between the polynomial and the ReLU function over a finite set of real-valued sample points $\{x_i\}_{i=1}^{N} \subset [-c, c]$, for some parameter $c > 0$:

$$\min_{a_0, \ldots, a_d} \sum_{i=1}^{N} \left( ReLU(x_i) - p_d(x_i) \right)^2$$

The resulting polynomial $p_d(x)$ provides a close approximation to ReLU within the interval $[-c, c]$. However, outside the interval, the polynomial behavior differs significantly from that of ReLU. ReLU exhibits piecewise linear growth:

$$ReLU(x) = \begin{cases} 0, & \text{if } x \leq 0 \\ x, & \text{if } x > 0 \end{cases} \quad \Rightarrow \quad ReLU(x) = \mathcal{O}(x)$$

In contrast, the growth of $p_d(x)$ for $d \geq 2$ is polynomial, dominated asymptotically by the highest-degree term:

$$p_d(x) = a_d x^d + a_{d-1} x^{d-1} + \cdots + a_1 x + a_0 = \mathcal{O}(x^d)$$

This fundamental mismatch in growth rates gives rise to the problem of *escaping activations*, first identified by Garimella et al. (2021). As inputs propagate through the layers of a neural network, the intermediate values passed into the polynomial activation can escape the intended interval $[-c, c]$, entering regions where $p_d(x)$ no longer approximates ReLU accurately. As a consequence, the network can produce excessively large activations, which in turn cause exploding weights and rapid degradation of model performance unless specific modifications are made to the training procedure to contain them.

## B.2 Coefficient Truncation

In privacy-preserving inference based on FHE, all computations are carried out using fixed-point arithmetic with limited precision. This constraint restricts the range and resolution of values that can be accurately represented, affecting both the domain of activation function inputs and the magnitudes of polynomial approximation coefficients. The polynomial coefficients $\{a_k\}_{k=0}^{d}$ are typically derived via floating-point least-squares fitting. To enable fixed-point evaluation under FHE, these coefficients are quantized using:

$$\tilde{a}_k = \frac{\lfloor a_k \cdot 2^b \rceil}{2^b}$$

where $b \in \mathbb{N}$ represents the number of fractional bits in the fixed-point format and $\lfloor \cdot \rceil$ denotes rounding to the nearest integer. The least-squares fitting procedure often produces small-magnitude coefficients, particularly for higher-degree monomials. If any coefficient satisfies $|a_k| < 2^{-(b+1)}$, the quantized value $\tilde{a}_k$ becomes zero. This effectively discards the corresponding monomial $x^k$ from the approximation. This phenomenon, referred to as *coefficient truncation* (Diaa et al., 2024), changes the shape of the polynomial and can cause significant deviation from the intended ReLU behavior, even within the designated approximation interval $[-c, c]$.

## C Why the Penalty Function Works?

**Lemma 1** (**Pre-activation Update Decomposition**). *Let $z_p^{(l)} = W^{(l)} h_{p_d}^{(l-1)}(x) \in \mathbb{R}^{n_l}$ denote the pre-activation vector at layer $l$ for input $x$. We define two quantities based on this vector: the clipping residual*

$$d^{(l)} = z_p^{(l)} - \text{clip}\left( z_p^{(l)}; [-c, c] \right)$$

which measures the amount by which $z_p^{(l)}$ exceeds the clipping range, and the gradient of the cross-entropy loss with respect to the pre-activation

$$g^{(l)} \;=\; \frac{\partial}{\partial z_p^{(l)}} \ell_{\mathrm{CE}}\left(g(x), y\right)$$

which captures the sensitivity of the loss to changes in $z_p^{(l)}$. After a single gradient-descent update with learning rate $\eta$ on the minibatch loss $\mathcal{L}_\mathcal{B}$, the change in $z_p^{(l)}$ decomposes as

$$\boxed{\Delta z_p^{(l)} = \underbrace{(-\eta)\left\|h_{p_d}^{(l-1)}(x)\right\|_2^2 g^{(l)}}_{\Delta z_{\mathrm{CE}}^{(l)}} + \underbrace{(-\eta)\,\zeta\left\|h_{p_d}^{(l-1)}(x)\right\|_2^2 \frac{d^{(l)}}{\left\|d^{(l)}\right\|_2}}_{\Delta z_{\mathrm{pen}}^{(l)}}}$$

Here, $\Delta z_{\mathrm{CE}}^{(l)}$ is the component induced by the cross-entropy loss, while $\Delta z_{\mathrm{pen}}^{(l)}$ is the component induced by the clip-range penalty.

**Proof.** Consider a single training example $(x, y)$, where $x$ is the input and $y$ the corresponding true label. The total per-sample loss includes two terms:

$$\ell(x) = \ell_{\mathrm{CE}}\left(g(x), y\right) \;+\; \zeta\, R\left(z_p^{(l)}\right), \quad R\left(z_p^{(l)}\right) \;=\; \left\|d^{(l)}\right\|_2$$

The first term is the standard cross-entropy loss computed from the network output $g(x)$. The second term is a layer-specific penalty that acts only on the pre-activations at layer $l$, penalizing values that fall outside the clipping interval. A gradient-descent step with learning rate $\eta$ updates the weight matrix by

$$\Delta W^{(l)} \;=\; -\eta\, \frac{\partial}{\partial W^{(l)}}\, \ell(x)$$

$$= -\eta\left(\frac{\partial}{\partial W^{(l)}}\, \ell_{\mathrm{CE}}\left(g(x), y\right) \;+\; \zeta\, \frac{\partial}{\partial W^{(l)}}\, R\left(z_p^{(l)}\right)\right)$$

Since $z_p^{(l)}$ depends linearly on the weights, its Jacobian with respect to $W^{(l)}$ is

$$\frac{\partial}{\partial W^{(l)}}\, z_p^{(l)} = \left[h_{p_d}^{(l-1)}(x)\right]^\top.$$

Using the chain rule, the gradient of the cross-entropy term with respect to the weights becomes

$$\frac{\partial}{\partial W^{(l)}}\, \ell_{\mathrm{CE}}\left(g(x), y\right) \;=\; \frac{\partial}{\partial z_p^{(l)}}\, \ell_{\mathrm{CE}}\left(g(x), y\right)\, \frac{\partial}{\partial W^{(l)}}\, z_p^{(l)}$$

$$= g^{(l)}\left[h_{p_d}^{(l-1)}(x)\right]^\top$$

The penalty function $R\left(z_p^{(l)}\right) \;=\; \left\|d^{(l)}\right\|_2$ is nonzero only when some elements lie outside $[-c, c]$. Its gradient with respect to $z_p^{(l)}$ is

$$\frac{\partial}{\partial z_p^{(l)}}\, R\left(z_p^{(l)}\right) \;=\; \begin{cases} \dfrac{d^{(l)}}{\left\|d^{(l)}\right\|_2} & \text{if } \left\|d^{(l)}\right\|_2 > 0, \\ 0 & \text{if } \left\|d^{(l)}\right\|_2 = 0. \end{cases}$$

This expression accounts for the case where the clipping has no effect; that is, when $z_p^{(l)} \in [-c, c]$ elementwise, the clipping residual $d^{(l)}$ becomes zero, and the gradient vanishes accordingly. Applying the chain rule again, we compute

$$\frac{\partial}{\partial W^{(l)}}\, R\left(z_p^{(l)}\right) \;=\; \frac{\partial}{\partial z_p^{(l)}}\, R\left(z_p^{(l)}\right)\, \frac{\partial}{\partial W^{(l)}}\, z_p^{(l)}$$

$$= \frac{d^{(l)}}{\left\|d^{(l)}\right\|_2}\left[h_{p_d}^{(l-1)}(x)\right]^\top$$

Putting it all together, we obtain the total gradient-based update for $W^{(l)}$

$$\Delta W^{(l)} = -\eta \left( g^{(l)} + \zeta \frac{d^{(l)}}{\left\| d^{(l)} \right\|_2} \right) \left[ h_{p_d}^{(l-1)}(x) \right]^\top$$

By definition, the change in pre-activation $z_p^{(l)}$ after one gradient descent step is given by

$$\begin{aligned}
\Delta z_p^{(l)} &= \overline{z}_p^{(l)} - z_p^{(l)} \\
&= \overline{W}^{(l)} h_{p_d}^{(l-1)}(x) - W^{(l)} h_{p_d}^{(l-1)}(x) \\
&= \left( W^{(l)} + \Delta W^{(l)} \right) h_{p_d}^{(l-1)}(x) - W^{(l)} h_{p_d}^{(l-1)}(x) \\
&= \Delta W^{(l)} h_{p_d}^{(l-1)}(x)
\end{aligned}$$

where $\overline{z}_p^{(l)}$ and $\overline{W}^{(l)}$ are updated pre-activations and weight matrix after one gradient-descent step. Substituting the previously derived expression for $\Delta W^{(l)}$

$$\begin{aligned}
\Delta z_p^{(l)} &= -\eta \left( g^{(l)} + \zeta \frac{d^{(l)}}{\left\| d^{(l)} \right\|_2} \right) \left[ h_{p_d}^{(l-1)}(x) \right]^\top h_{p_d}^{(l-1)}(x) \\
&= -\eta \left( g^{(l)} + \zeta \frac{d^{(l)}}{\left\| d^{(l)} \right\|_2} \right) \left\| h_{p_d}^{(l-1)}(x) \right\|_2^2 \\
&= \underbrace{(-\eta) \left\| h_{p_d}^{(l-1)}(x) \right\|_2^2 g^{(l)}}_{\Delta z_{\text{CE}}^{(l)}} + \underbrace{(-\eta)\,\zeta \left\| h_{p_d}^{(l-1)}(x) \right\|_2^2 \frac{d^{(l)}}{\left\| d^{(l)} \right\|_2}}_{\Delta z_{\text{pen}}^{(l)}}
\end{aligned}$$

we obtain the penalty components $\Delta z_{\text{CE}}^{(l)}$ and $\Delta z_{\text{pen}}^{(l)}$ as mentioned in the lemma. ∎

**Lemma 2** (**Clipping Gradient Pullback**). *Let $\Delta z_{\text{pen}}^{(l)}$ be the penalty-induced component of the pre-activation update from Lemma 1, and $d^{(l)}$ be the clipping residual. Then the inner product between $\Delta z_{\text{pen}}^{(l)}$ and $d^{(l)}$ satisfies*

$$\boxed{\left\langle \Delta z_{\text{pen}}^{(l)}, d^{(l)} \right\rangle = -\eta\,\zeta \left\| h_{p_d}^{(l-1)}(x) \right\|_2^2 \left\| d^{(l)} \right\|_2 < 0}$$

*whenever $\left\| d^{(l)} \right\|_2 \neq 0$. Therefore, $\Delta z_{\text{pen}}^{(l)}$ "pulls back" each element of $z_p^{(l)}$ lying outside $[-c, c]$ towards the clipping interval.*

***Proof.*** The clipping residual $d^{(l)}$ is defined as the difference between the current pre-activation $z_p^{(l)}$ and its clipped version. $\Delta z_{\text{pen}}^{(l)}$ is the component in $\Delta z_p^{(l)}$ induced by the clip-range penalty. Taking the inner product of $\Delta z_{\text{pen}}^{(l)}$ with the clipping residual $d^{(l)}$ gives

$$\begin{aligned}
\left\langle \Delta z_{\text{pen}}, d^{(l)} \right\rangle &= \left\langle -\eta\,\zeta \left\| h_{p_d}^{(l-1)}(x) \right\|_2^2 \frac{d^{(l)}}{\left\| d^{(l)} \right\|_2}, d^{(l)} \right\rangle \\
&= -\eta\,\zeta \left\| h_{p_d}^{(l-1)}(x) \right\|_2^2 \left\langle \frac{d^{(l)}}{\left\| d^{(l)} \right\|_2}, d^{(l)} \right\rangle \\
&= -\eta\,\zeta \left\| h_{p_d}^{(l-1)}(x) \right\|_2^2 \frac{1}{\left\| d^{(l)} \right\|_2} \langle d^{(l)}, d^{(l)} \rangle \\
&= -\eta\,\zeta \left\| h_{p_d}^{(l-1)}(x) \right\|_2^2 \frac{1}{\left\| d^{(l)} \right\|_2} \left\| d^{(l)} \right\|_2^2 \\
&= -\eta\,\zeta \left\| h_{p_d}^{(l-1)}(x) \right\|_2^2 \left\| d^{(l)} \right\|_2
\end{aligned}$$

Each factor on the right-hand side: $\eta$ (learning rate), $\zeta$ (regularization parameter), $\left\|h_{p_d}^{(l-1)}(x)\right\|_2^2$ (squared norm of the preceding hidden representation) and $\left\|d^{(l)}\right\|_2$ (magnitude of the clipping residual) is strictly positive whenever any pre-activation lies outside $[-c, c]$. Hence the entire product is strictly negative

$$\left\langle \Delta z_{\text{pen}}^{(l)}, \ d^{(l)} \right\rangle \ < \ 0$$

A negative inner product implies that $\Delta z_{\text{pen}}^{(l)}$ points in the exact opposite direction of the clipping residual $d^{(l)}$. Consequently, every out-of-range component of pre-activation $z_p^{(l)}$ is pulled back towards the clipping boundary, while pre-activation already within $[-c, c]$ experience no change. ∎

## D  TRAINING STABILITY DURING POLYNOMIAL TRAINING

### D.1  PRE-ACTIVATION CLIPPING

During early training stages, network weights remain close to their random initialization, and the optimization process does not immediately constrain the pre-activation values to lie within the target approximation interval $[-c, c]$. As a result, unbounded pre-activations may arise before the model learns to keep them within range, leading to training instability that can degrade model behavior beyond recovery. To prevent this, we restrict pre-activations to the interval $[-c, c]$ before evaluating the polynomial activation, which is achieved through a *clipping strategy* $\text{clip}(z; [-c, c])$ (Diaa et al., 2024):

$$h_{p_d}^{(l)}(x) \ = \ p_d \left( \text{clip} \left( W^{(l)} h_{p_d}^{(l-1)}(x); [-c, c] \right) \right).$$

Importantly, this *clipping operation is applied only during training* to stabilize learning. It is performed after computing the clip-range penalty to ensure that gradients from the regularization term, which encourages the network to keep pre-activations within the approximation interval $[-c, c]$, are preserved and not masked by the clipping. At inference time, the clipping function is removed:

$$h_{p_d}^{(l)}(x) \ = \ p_d \left( W^{(l)} h_{p_d}^{(l-1)}(x) \right).$$

The model, having learned to constrain pre-activations during training, is expected to remain within the approximation interval without explicit clipping at inference.

### D.2  REGULARIZATION WARM-UP

While activation regularization and pre-activation clipping are both necessary for training stability, applying the full regularization strength from the outset can lead to numerical instability, particularly in larger models. In the early epochs, many pre-activation values lie outside the target interval $[-c, c]$ across several layers. The clip-range penalty adds a contribution from every such layer, so the penalty term becomes very large. In extreme cases, this can cause the total loss to become numerically undefined.

To address this issue, the regularization strength $\zeta$ is progressively increased over the initial training epochs, which is implemented through a *regularization warm up schedule* (Diaa et al., 2024). Let $T_{warm}$ denote the total number of warm-up epochs. For each epoch $t$, we define a *time-dependent regularization weight* $\zeta_t$ as:

$$\zeta_t \ = \ \begin{cases} \alpha_t \cdot \zeta & \text{if } t \leq T_{warm} \\ \zeta & \text{if } t > T_{warm} \end{cases},$$

where $\{\alpha_t\}_{t=1}^{T_{warm}}$ is a fixed sequence of scaling factors satisfying $0 < \alpha_1 < \cdots < \alpha_{T_{warm}} < 1$. In practice, we empirically find that setting $T_{warm} = 4$ provides a stable convergence. We use a simple predefined sequence of scaling factors: $\alpha = \left\{ \frac{1}{100}, \frac{1}{50}, \frac{1}{10}, \frac{1}{5} \right\}$, which produces the epoch-wise regularization strength: $\zeta_t \in \left\{ \frac{\zeta}{100}, \frac{\zeta}{50}, \frac{\zeta}{10}, \frac{\zeta}{5}, \zeta, \zeta, \ldots \right\}$. This warm-up schedule ensures that the regularization penalty is introduced progressively in the initial training epochs without suffering from exploding loss values.

# E ADDITIONAL DETAILS ON STRUCTURAL OPTIMIZATIONS

## E.1 DETAILED DERIVATIONS OF NODE FUSING

Batch normalization in linear form is given by

$$B(x) = b_1 x + b_0, \quad b_1 = \frac{\gamma}{\sigma}, \quad b_0 = \beta_b - b_1 \mu.$$

A quadratic activation $P(x)$ serves as a polynomial substitute for ReLU, where $P(x)$ is given by

$$P(x) = c_2 x^2 + c_1 x + c_0.$$

**Case 1:** $P(B(x)) \mapsto P(x)$. Start with $B(x) = b_1 x + b_0$.

$$\begin{aligned}
P(B(x)) &= c_2 \big(B(x)\big)^2 + c_1 B(x) + c_0 \\
&= c_2 (b_1 x + b_0)^2 + c_1 (b_1 x + b_0) + c_0 \\
&= c_2 \big(b_1^2 x^2 + 2 b_1 b_0 x + b_0^2\big) + c_1 b_1 x + c_1 b_0 + c_0 \\
&= (c_2 b_1^2) x^2 + \big(2 c_2 b_1 b_0 + c_1 b_1\big) x + \big(c_2 b_0^2 + c_1 b_0 + c_0\big).
\end{aligned}$$

Resulting polynomial coefficients

$$p_2 = b_1^2 c_2, \quad p_1 = b_1 (2 b_0 c_2 + c_1), \quad p_0 = b_0^2 c_2 + b_0 c_1 + c_0.$$

**Case 2:** $B(C(x)) \mapsto C(x)$. Consider a convolution $C(x) = \sum_i w_i x_i + \beta_c$. When batch normalization is applied, it can be rewritten as a rescaled convolution with updated weights and bias.

$$\begin{aligned}
B(C(x)) &= b_1 C(x) + b_0 \\
&= b_1 \Big( \sum_i w_i x_i + \beta_c \Big) + b_0 \\
&= \sum_i (b_1 w_i) x_i + (b_1 \beta_c + b_0).
\end{aligned}$$

The equivalent convolution $\sum_i \omega_i x_i + \alpha$ uses the following updated parameters

$$\omega_i = b_1 w_i, \quad \alpha = b_1 \beta_c + b_0.$$

**Case 3:** $P(B_X(x) + B_Y(y)) \mapsto S(x, y)$. Define $B_X(x) = b_{X1} x + b_{X0}$ and $B_Y(y) = b_{Y1} y + b_{Y0}$. Each of these represents a linearized batch normalization applied to one input variable. Let $z = B_X(x) + B_Y(y) = b_{X1} x + b_{Y1} y + (b_{X0} + b_{Y0})$. Applying the activation

$$\begin{aligned}
P(z) &= c_2 z^2 + c_1 z + c_0 \\
&= c_2 \big(b_{X1} x + b_{Y1} y + b_{X0} + b_{Y0}\big)^2 + c_1 \big(b_{X1} x + b_{Y1} y + b_{X0} + b_{Y0}\big) + c_0.
\end{aligned}$$

Expanding the square term

$$z^2 = b_{X1}^2 x^2 + b_{Y1}^2 y^2 + 2 b_{X1} b_{Y1} xy + 2 b_{X1}(b_{X0} + b_{Y0}) x + 2 b_{Y1}(b_{X0} + b_{Y0}) y + (b_{X0} + b_{Y0})^2.$$

Substituting and grouping terms by monomials

$$\begin{aligned}
P(z) = \underbrace{c_2 b_{X1}^2}_{d_{X2}} x^2 + \underbrace{c_2 b_{Y1}^2}_{d_{Y2}} y^2 + \underbrace{2 c_2 b_{X1} b_{Y1}}_{d_{XY}} xy \\
+ \underbrace{\big[b_{X1}\big(2 c_2 (b_{X0} + b_{Y0}) + c_1\big)\big]}_{d_X} x + \underbrace{\big[b_{Y1}\big(2 c_2 (b_{X0} + b_{Y0}) + c_1\big)\big]}_{d_Y} y \\
+ \underbrace{\big[c_2 (b_{X0} + b_{Y0})^2 + c_1 (b_{X0} + b_{Y0}) + c_0\big]}_{d_0}.
\end{aligned}$$

Hence $S(x, y) = d_{X2} x^2 + d_{Y2} y^2 + d_{XY} xy + d_X x + d_Y y + d_0$ with the coefficients above.

**Case 4:** $P(B_X(x) + y) \mapsto S(x, y)$. This is the identity shortcut case. Set $B_X(x) = b_{X1}x + b_{X0}$ and define the combined input as $z = B_X(x) + y = b_{X1}x + y + b_{X0}$. Applying the activation

$$P(z) = c_2 z^2 + c_1 z + c_0.$$

Expanding the square term

$$z^2 = b_{X1}^2 x^2 + y^2 + 2b_{X1}xy + 2b_{X1}b_{X0}x + 2b_{X0}y + b_{X0}^2.$$

Substituting and grouping terms by monomials

$$P(z) = \underbrace{c_2 b_{X1}^2}_{d_{X2}} x^2 + \underbrace{c_2}_{d_{Y2}} y^2 + \underbrace{2c_2 b_{X1}}_{d_{XY}} xy + \underbrace{\left[ b_{X1}(2c_2 b_{X0} + c_1) \right]}_{d_X} x$$

$$+ \underbrace{\left[ 2c_2 b_{X0} + c_1 \right]}_{d_Y} y + \underbrace{\left[ c_2 b_{X0}^2 + c_1 b_{X0} + c_0 \right]}_{d_0}.$$

Hence $S(x, y) = d_{X2}x^2 + d_{Y2}y^2 + d_{XY}xy + d_X x + d_Y y + d_0$ with the coefficients above.

### E.2 DETAILED DERIVATIONS OF WEIGHT REDISTRIBUTION

#### E.2.1 UPDATE FORWARD

**Donors.** *Average Pooling.* Start with $\mu(x) = k^{-1} \sum_i x_i$ and its normalized form $\bar{\mu}(x) = \sum_i x_i$. We must find $\upsilon$ such that $\upsilon \bar{\mu}(x) = \mu(x)$ is valid

$$\upsilon \bar{\mu}(x) = \mu(x)$$

$$\upsilon \sum_i x_i = k^{-1} \sum_i x_i$$

$$\upsilon = k^{-1}.$$

*Polynomial Functions.* Let $P(x) = \sum_{i=0}^{d} c_i x^i$ and $\bar{P}(x) = x^d + \sum_{i=0}^{d-1} \bar{c}_i x^i$ be a degree-$d$ polynomial and its normalization, respectively. For $\upsilon \bar{P}(x) = P(x)$ to hold we have $\upsilon$ given by

$$\upsilon \bar{P}(x) = P(x)$$

$$\upsilon \left( x^d + \sum_{i=0}^{d-1} \bar{c}_i x^i \right) = \sum_{i=0}^{d} c_i x^i$$

$$\upsilon x^d = c_d x^d$$

$$\upsilon = c_d.$$

The normalized coefficients must satisfy

$$\upsilon \bar{P}(x) = P(x)$$

$$\upsilon \left( x^d + \sum_{i=0}^{d-1} \bar{c}_i x^i \right) = \sum_{i=0}^{d} c_i x^i$$

$$\upsilon \bar{c}_i x^i = c_i x^i$$

$$\bar{c}_i = c_i \upsilon^{-1} \quad \forall i \in \{0, \dots, d-1\}.$$

*Bivariate Polynomial.* Consider the degree-$d$ bivariate polynomial $S(x, y) = \sum_{i=0}^{d} \sum_{j=0}^{d-i} c_{ij} x^i y^j$, and let $\bar{S}(x, y) = x^d + \sum_{i=0}^{d-1} \sum_{j=0}^{d-i} \bar{c}_{ij} x^i y^j$ be its normalization on $x$ (normalizing on $y$ is equivalent). As in the univariate case, we require $\upsilon \bar{S}(x, y) = S(x, y)$. Thus, $\upsilon$ is determined by

$$\upsilon \bar{S}(x, y) = S(x, y)$$

$$\upsilon \left( x^d + \sum_{i=0}^{d-1} \sum_{j=0}^{d-i} \bar{c}_{ij} x^i y^j \right) = \sum_{i=0}^{d} \sum_{j=0}^{d-i} c_{ij} x^i y^j$$

$$\upsilon x^d = c_{d0} x^d$$

$$\upsilon = c_{d0}.$$

The normalized coefficients are obtained through

$$v\bar{S}(x,y) = S(x,y)$$

$$v\left(x^d + \sum_{i=0}^{d-1}\sum_{j=0}^{d-i}\bar{c}_{ij}x^iy^j\right) = \sum_{i=0}^{d}\sum_{j=0}^{d-i}c_{ij}x^iy^j$$

$$v\bar{c}_{ij}x^iy^j = c_{ij}x^iy^j$$

$$\bar{c}_{ij} = c_{ij}v^{-1} \quad \forall i \in \{0,\ldots,d-1\}, \forall j \in \{0,\ldots,d-i\}.$$

**Receivers.** *Kernel Functions.* Let $K(x) = \sum_{i\in\mathcal{I}} w_i x_i + \beta$ be the original kernel function and $\bar{K}(x) = \sum_{i\in\mathcal{I}} \bar{w}_i x_i + \bar{\beta}$ its updated version. To ensure model equivalence, it must hold that $\bar{K}(v^{-1}x) = K(x)$. Therefore, the updated parameters are given as follows

$$\bar{K}(v^{-1}x) = K(x)$$

$$\sum_{i\in\mathcal{I}} \bar{w}_i v^{-1} x_i + \bar{\beta} = \sum_{i\in\mathcal{I}} w_i x_i + \beta$$

$$\bar{w}_i v^{-1} x_i = w_i x_i$$

$$\bar{w}_i = w_i v \quad \forall i \in \mathcal{I}, \quad \bar{\beta} = \beta.$$

*Polynomial Functions.* For a degree-$d$ polynomial receiver $P(x) = \sum_{i=0}^{d} c_i x^i$, the coefficients of its updated version $\bar{P}(x) = \sum_{i=0}^{d} \bar{c}_i x^i$ must be chosen such that $\bar{P}(v^{-1}x) = P(x)$ holds

$$\bar{P}(v^{-1}x) = P(x)$$

$$\sum_{i=0}^{d} \bar{c}_i (v^{-1}x)^i = \sum_{i=0}^{d} c_i x^i$$

$$\bar{c}_i v^{-i} x^i = c_i x^i$$

$$\bar{c}_i = c_i v^i \quad \forall i \in \{0,\ldots,d\}.$$

*Bivariate Polynomial.* For a degree-$d$ bivariate polynomial $S(x,y) = \sum_{i=0}^{d}\sum_{j=0}^{d-i} c_{ij}x^iy^j$ and its updated version $\bar{S}(x,y) = \sum_{i=0}^{d}\sum_{j=0}^{d-i} \bar{c}_{ij}x^iy^j$,, normalizing with respect to $x$ requires that $\bar{S}(v^{-1}x,y) = S(x,y)$. Consequently, the coefficients of $\bar{S}(\cdot)$ are

$$\bar{S}(v^{-1}x,y) = S(x,y)$$

$$\sum_{i=0}^{d}\sum_{j=0}^{d-i} \bar{c}_{ij}(v^{-1}x)^iy^j = \sum_{i=0}^{d}\sum_{j=0}^{d-i} c_{ij}x^iy^j$$

$$\bar{c}_{ij}v^{-i}x^iy^j = c_{ij}x^iy^j$$

$$\bar{c}_{ij} = c_{ij}v^i \quad \forall i \in \{0,\ldots,d\}, \forall j \in \{0,\ldots,d-i\}.$$

### E.2.2 UPDATE BACKWARD

**Donors.** *Average Pooling.* Start with $\mu(x) = k^{-1}\sum_i x_i$ and its normalization $\bar{\mu}(x) = \sum_i x_i$. We must determine $v$ such that $\bar{\mu}(vx) = \mu(x)$ holds

$$\bar{\mu}(vx) = \mu(x)$$

$$\sum_i vx_i = k^{-1}\sum_i x_i$$

$$v\sum_i x_i = k^{-1}\sum_i x_i$$

$$v = k^{-1}.$$

*Polynomial Functions.* For a degree-$d$ polynomial donor $P(\cdot)$, the equality $\bar{P}(\upsilon x) = P(x)$ must be satisfied, where $\bar{P}(\cdot)$ is the normalized version. The appropriate $\upsilon$ is found from

$$\bar{P}(\upsilon x) = P(x)$$

$$(\upsilon x)^d + \sum_{i=0}^{d-1} \bar{c}_i (\upsilon x)^i = \sum_{i=0}^{d} c_i x^i$$

$$\upsilon^d x^d = c_d x^d$$

$$\upsilon = c_d^{1/d}.$$

and the normalized coefficients are

$$\bar{P}(\upsilon x) = P(x)$$

$$(\upsilon x)^d + \sum_{i=0}^{d-1} \bar{c}_i (\upsilon x)^i = \sum_{i=0}^{d} c_i x^i$$

$$\bar{c}_i \upsilon^i x^i = c_i x^i$$

$$\bar{c}_i = c_i \upsilon^{-i} \quad \forall i \in \{0, \ldots, d-1\}.$$

*Bivariate Polynomial.* Assuming normalization along $x$, we require $\bar{S}(\upsilon x, y) = S(x, y)$ for the degree-$d$ bivariate polynomial $S(x, y)$ and its normalization $\bar{S}(x, y)$. For that, $\upsilon$ is determined from

$$\bar{S}(\upsilon x, y) = S(x, y)$$

$$(\upsilon x)^d + \sum_{i=0}^{d-1} \sum_{j=0}^{d-i} \bar{c}_{ij} (\upsilon x)^i y^j = \sum_{i=0}^{d} \sum_{j=0}^{d-i} c_{ij} x^i y^j$$

$$\upsilon^d x^d = c_{d0} x^d$$

$$\upsilon = c_{d0}^{1/d}.$$

with the normalized coefficients being

$$\bar{S}(\upsilon x, y) = S(x, y)$$

$$(\upsilon x)^d + \sum_{i=0}^{d-1} \sum_{j=0}^{d-i} \bar{c}_{ij} (\upsilon x)^i y^j = \sum_{i=0}^{d} \sum_{j=0}^{d-i} c_{ij} x^i y^j$$

$$\bar{c}_{ij} \upsilon^i x^i y^j = c_{ij} x^i y^j$$

$$\bar{c}_{ij} = c_{ij} \upsilon^{-i} \quad \forall i \in \{0, \ldots, d-1\}, \forall j \in \{0, \ldots, d-i\}.$$

**Receivers.** *Kernel Functions.* Let $K(x) = \sum_{i \in \mathcal{I}} w_i x_i + \beta$ and $\bar{K}(x) = \sum_{i \in \mathcal{I}} \bar{w}_i x_i + \bar{\beta}$ be a kernel function and its updated version. To preserve model equivalence, we require $\bar{K}(x) = \upsilon K(x)$, thus

$$\bar{K}(x) = \upsilon K(x)$$

$$\sum_{i \in \mathcal{I}} \bar{w}_i x_i + \bar{\beta} = \upsilon \Big( \sum_{i \in \mathcal{I}} w_i x_i + \beta \Big)$$

$$\bar{w}_i x_i = \upsilon w_i x_i$$

$$\bar{w}_i = w_i \upsilon \quad \forall i \in \mathcal{I}, \quad \bar{\beta} = \beta \upsilon.$$

*Polynomial Functions.* For a degree-$d$ polynomial receiver $P(x) = \sum_{i=0}^{d} c_i x^i$, the updated polynomial $\bar{P}(x) = \sum_{i=0}^{d} \bar{c}_i x^i$ must satisfy $\bar{P}(x) = \upsilon P(x)$. This yields

$$\bar{P}(x) = \upsilon P(x)$$

$$\sum_{i=0}^{d} \bar{c}_i x^i = \upsilon \sum_{i=0}^{d} c_i x^i$$

$$\bar{c}_i x^i = \upsilon c_i x^i$$

$$\bar{c}_i = c_i \upsilon \quad \forall i \in \{0, \ldots, d\}.$$

*Bivariate Polynomial.* For a degree-$d$ bivariate polynomial $S(x, y) = \sum_{i=0}^{d} \sum_{j=0}^{d-i} c_{ij} x^i y^j$ and its updated version $\bar{S}(x, y) = \sum_{i=0}^{d} \sum_{j=0}^{d-i} \bar{c}_{ij} x^i y^j$, we require $\bar{S}(x, y) = \upsilon S(x, y)$. Therefore

$$\bar{S}(x, y) = \upsilon S(x, y)$$

$$\sum_{i=0}^{d} \sum_{j=0}^{d-i} \bar{c}_{ij} x^i y^j = \upsilon \sum_{i=0}^{d} \sum_{j=0}^{d-i} c_{ij} x^i y^j$$

$$\bar{c}_{ij} x^i y^j = \upsilon c_{ij} x^i y^j$$

$$\bar{c}_{ij} = c_{ij} \upsilon \quad \forall i \in \{0, \ldots, d\}, \forall j \in \{0, \ldots, d-i\}.$$

### E.3 EXAMPLE OF TOWER REUSE

Suppose following an activation function, there is a convolution. Let modulus $q_i \approx \Delta^2$, thus $\lambda(q_i) = 2$. Consider an input $x$ with scaling factor $\Delta_x = \Delta$, giving $\lambda(x) = 1$. Applying a quadratic activation $y = x^2 + c_1 x + c_0$ yields $\lambda(y) = 2$, with $\lambda(c_1) = 1$ and $\lambda(c_0) = 2$. Next, a convolution $z = \sum_i \omega_i y_i + \alpha$ is performed. Since $\lambda(y) = 2$, by setting $\lambda(\omega) = 1$, we obtain $\lambda(z) = 3$. As $\lambda(z) > \lambda(q_i)$, a rescale is triggered. The rescale operation drops $q_i$ from the modulus chain, reducing the level $\Lambda(\cdot)$ by one and resetting the sublevel to one:

$$\Lambda(\downarrow z) = \Lambda(z) - 1, \qquad \lambda(\downarrow z) = \lambda(z) - \lambda(q_i) = 1.$$

In terms of $\Delta$, a ciphertext $z$ with $\Delta_z = \Delta^3$ rescaled by $q_i \approx \Delta^2$ results in output scale $\Delta_{\downarrow z} \approx \Delta$.

### E.4 LEVEL ANALYSIS

Table 1: Number of levels required for multiplications in ResNet models. $P_4$: Degree-4 polynomial activation used in PILLAR (Diaa et al., 2024). $P_2$: Degree-2 polynomial activation (§2). $P_2F$: $P_2$ with node fusing (§3.1). $P_2R$: $P_2$ with weight redistribution (§3.2). $P_2FR$: $P_2$ with node fusing and weight redistribution. $P_2FRT$: $P_2FR$ with tower reuse (§3.3).

| **Model** | $P_4$ | $P_2$ | $P_2F$ | $P_2R$ | $P_2FR$ | $P_2FRT$ |
|---|---|---|---|---|---|---|
| ResNet-18 | 87 | 70 | 53 | 35 | 35 | 18 |
| ResNet-20 | 97 | 78 | 59 | 39 | 39 | 20 |
| ResNet-32 | 157 | 126 | 95 | 63 | 63 | 32 |

Table 1 summarizes the number of levels $\mathcal{L}$ required for multiplication across different ResNet variants under various optimization techniques. To illustrate, we focus on ResNet-18, which comprises 17 convolutional layers, batch normalization layers, and activation functions, along with a single average pooling and linear layer:

- $P_4$: Prior work achieving the lowest $\mathcal{L}$ (Diaa et al., 2024) uses a degree-4 polynomial as activation function. This results in $\mathcal{L} = 87$, with each convolution, batch normalization, average pooling, and linear layer contributing $\mathcal{L} = 1$, and the polynomial activation $\mathcal{L} = 3$.

- $P_2$: By replacing the activation function with a degree-2 polynomial, as described in §2, the required level for activations drops to 2, reducing the model $\mathcal{L}$ to 70.

- $P_2F$: Applying node fusing (§3.1) to $P_2$ removes all batch normalizations, lowering $\mathcal{L}$ to 53.

- $P_2R$: Alternatively, applying weight redistribution (§3.2) to $P_2$ yields $\mathcal{L} = 35$. This is achieved because the highest-order coefficients in batch normalization and polynomial activations are normalized to one, simplifying them to $x + b_0$ ($\mathcal{L} = 0$) and $x^2 + c_1 x + c_0$ ($\mathcal{L} = 1$), respectively. Moreover, the divisor in average pooling is also normalized to one, giving it $\mathcal{L} = 0$. These simplifications collectively result in $\mathcal{L} = 35$.

- $P_2FR$: Combining node fusing and weight redistribution retains $\mathcal{L} = 35$. However, it requires fewer homomorphic operations overall, since node fusing eliminates batch normalizations entirely.

- $P_2FRT$: Finally, applying tower reuse (§3.3) to $P_2FR$ allows a non-linear polynomial activation followed by a linear kernel function to share the same level. This halves the total levels to $\mathcal{L} = 18$, following $\mathcal{L}_{P_2FRT} = \lceil \frac{\mathcal{L}_{P_2FR}}{2} \rceil$ for ResNet models.

In summary, our techniques reduce levels by nearly a factor of five compared to prior work.

## F ILLUSTRATION OF SLICE-WISE CLUSTERING

### F.1 SINGLE MODEL SLICE CLUSTERING

Figure 3 illustrates how convolution filters are divided into slices along the width axis $W_l$, with each colored slice $S_s^{(l)}$ clustered independently using its own codebook. Each 3D block in the figure corresponds to a convolution filter associated with one output channel. The three axes correspond to kernel height $H_l$ (vertical), kernel width $W_l$ (horizontal), and input channels $I_l$ (depth). The collection of such filters forms the output channels $O_l$. Slices are taken as vertical strips along the width axis $W_l$, and the figure highlights them with distinct colors. The blue strip corresponds to slice $S_1^{(l)}$, the yellow strip to slice $S_2^{(l)}$, and the pink strip to slice $S_3^{(l)}$. Each slice $S_s^{(l)}$ groups together all weights $W_{o,i,h,s}^{(l)}$ across output channels $O_l$, input channels $I_l$, and kernel height $H_l$ at a fixed width position $s$, and is clustered independently with its own codebook $\mathcal{C}_s^{(l)}$. This illustration emphasizes that clustering is performed slice by slice, rather than across the entire filter volume.

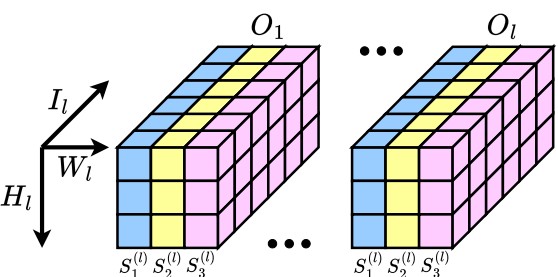

Figure 3: Illustration of single model slice-wise clustering.

### F.2 ENSEMBLE SLICE CLUSTERING

Figure 4 illustrates how slice-wise clustering is extended from a single model to an ensemble of polynomial networks. Each 3D block corresponds to the convolution filters of one model instance, spanning kernel height $H_l$ (vertical), kernel width $W_l$ (horizontal), and input channels $I_l$ (depth). The collection of such filters across output channels $O_l$ forms the layer weights $W^{(l,m)}$ for model $m$. For a fixed layer $l$, slices are again taken as vertical strips along the width axis $W_l$, with colors indicating different slices $S_s^{(l,m)}$ within each model. Unlike the single-model case, however, we now align slices from multiple models at the same width position $s$ and stack them together. This produces a slice matrix $X_s^{(l)} \in \mathbb{R}^{N_s \times M}$, where each row corresponds to one weight coordinate across all $M$ models, and $N_s = O_l I_l H_l$. Clustering is performed jointly in $\mathbb{R}^M$, producing a shared codebook $\mathcal{C}_s^{(l)}$ for slice $s$. The illustration emphasizes that ensemble clustering is not applied to each model independently. Instead, slices are stacked across models at the same spatial position, and clustering is performed jointly.

## G ABLATION STUDY

### G.1 ABLATION ON CLIPPING RANGE (VALUE OF $c$)

We conduct an ablation study to evaluate the effect of the clipping interval $[-c, c]$ on model performance. The choice of $c$ determines how much of the activation distribution is preserved versus truncated: if $c$ is too small, informative activations are excessively clipped, which can harm representational capacity, while if $c$ is too large, the clipping mechanism provides little stabilization benefit, allowing unstable activations to propagate. Figure 5 shows the results for CIFAR-10 with ResNet-18. Figure 5a presents the distribution of pre-activation inputs to ReLU across the entire network, where the values are concentrated within the interval $[-2, 2]$ and only a small fraction lies outside this range, supporting the choice of restricting pre-activations to $[-2, 2]$. Figure 5b reports the accuracy obtained under different clipping intervals. While several ranges were tested, $[-2, 2]$ consistently achieved the highest mean accuracy across runs. Smaller intervals degraded performance by removing too many activations, whereas larger intervals reduced stability and resulted in lower accuracy. The same trend holds across other dataset–architecture combinations. Based on these observations, we adopt $[-2, 2]$ as the clipping range in all our experiments.

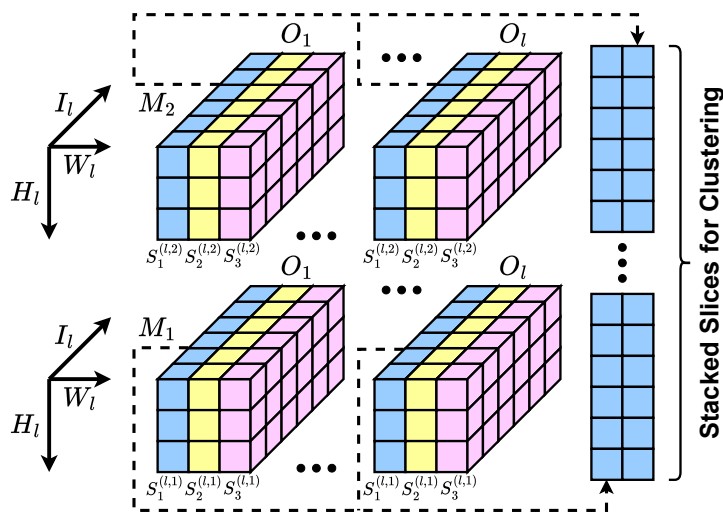

Figure 4: Illustration of ensemble slice-wise clustering.

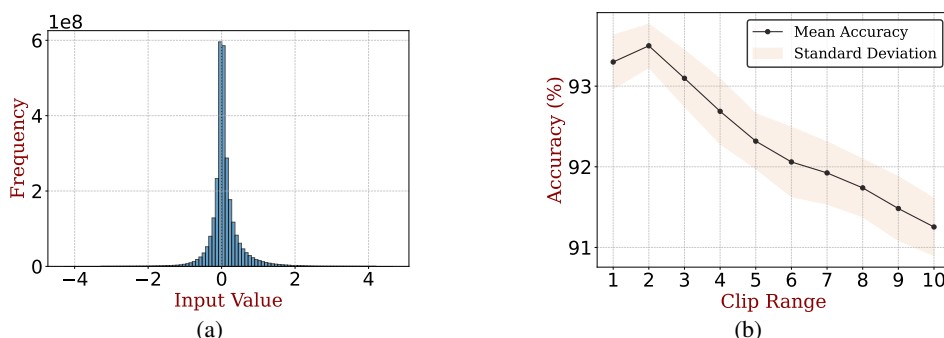

Figure 5: Results of the clipping range ablation study on CIFAR-10 with ResNet-18. (a) Distribution of pre-activation inputs to ReLU across the entire network, showing concentration within $[-2, 2]$. (b) Test accuracy under different clipping intervals, where $[-2, 2]$ produces the best performance.

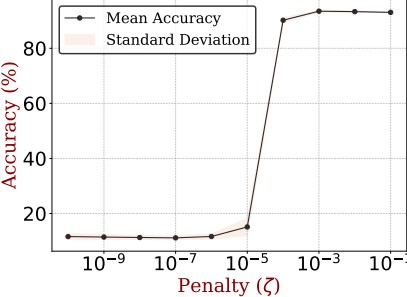

Figure 6: Results of the penalty strength ablation study on CIFAR-10 with ResNet-18, showing test accuracy across different values of $\zeta$. The best performance is achieved at $\zeta = 10^{-3}$; smaller values fail to regularize effectively, while larger values start to decrease accuracy.

### G.2 ABLATION ON PENALTY STRENGTH (VALUE OF $\zeta$)

We conduct an ablation study to evaluate the effect of the penalty strength $\zeta$ on model performance. The parameter $\zeta$ controls the magnitude of the regularization term that penalizes pre-activations outside the target interval $[-c, c]$. Figure 6 shows the results for CIFAR-10 with ResNet-18. The highest accuracy is obtained for $\zeta = 10^{-3}$. Values smaller than $10^{-3}$ do not provide sufficient regularization and result in poor accuracy, while values greater than $10^{-3}$ start decreasing performance, indicating that excessive penalization constrains the model too strongly. The same trend holds across other dataset–architecture combinations. Based on these results, we set $\zeta = 10^{-3}$ in all our experiments.

### G.3 COMPARISON WITH PILLAR

We evaluate PILLAR using its open-source implementation (git, 2024b). PILLAR is designed for interactive PPML, which is a different use case from our non-interactive setting, and its repository does not provide code for private inference. For these reasons, we compare only in terms of accuracy and do not evaluate inference time. Table 2 reports accuracies (mean $\pm$ standard deviation) for ResNet-18, ResNet-20, and ResNet-32 on CIFAR-10 and CIFAR-100 using ReLU, PAPER (our method), and two versions of PILLAR. The original PILLAR is defined with a degree-4 polynomial. For a fair comparison, we additionally adapted this implementation to use a degree-2 polynomial, allowing a direct comparison to PAPER. On average, the adapted degree-2 version shows an accuracy drop of 6.5% with very high variance, particularly on CIFAR-100. The original degree-4 version is more stable but still falls short of ReLU by about 5.4% on average. In contrast, PAPER with degree-2 consistently stays much closer to ReLU, with only about a 1.3% drop on average. Importantly, PAPER remains stable across both datasets and all ResNet variants, avoiding the large fluctuations seen in PILLAR. These results demonstrate that while PAPER can exploit low-degree polynomials effectively, PILLAR either collapses when reduced to degree 2 or, in its intended degree-4 form, does not reach comparable accuracy.

Table 2: Classification accuracies (mean $\pm$ standard deviation) on CIFAR-10 and CIFAR-100 for ResNet-18, ResNet-20, and ResNet-32 using ReLU, PAPER (degree-2), and two versions of PILLAR. The original PILLAR method is degree-4 (Diaa et al., 2024), while the degree-2 variant is our adaptation of its open-source implementation for comparison.

| Model | Dataset | ReLU | PAPER | PILLAR ($d=2$) | PILLAR ($d=4$) |
|---|---|---|---|---|---|
| ResNet-18 | CIFAR-10 | $94.57 \pm 0.23$ | $93.73 \pm 0.27$ | $91.78 \pm 0.13$ | $93.26 \pm 0.15$ |
| | CIFAR-100 | $76.22 \pm 0.44$ | $74.96 \pm 0.51$ | $71.92 \pm 0.28$ | $74.81 \pm 0.22$ |
| ResNet-20 | CIFAR-10 | $94.84 \pm 0.25$ | $94.06 \pm 0.26$ | $90.43 \pm 0.40$ | $90.98 \pm 0.29$ |
| | CIFAR-100 | $74.96 \pm 0.36$ | $73.89 \pm 0.37$ | $62.30 \pm 3.96$ | $62.55 \pm 7.51$ |
| ResNet-32 | CIFAR-10 | $95.23 \pm 0.30$ | $93.86 \pm 0.24$ | $90.40 \pm 1.73$ | $92.24 \pm 0.27$ |
| | CIFAR-100 | $76.58 \pm 0.40$ | $73.93 \pm 0.53$ | $66.38 \pm 1.78$ | $66.10 \pm 1.12$ |

# H ADDITIONAL EXPERIMENTAL RESULTS

## H.1 ADDITIONAL DETAILS OF FHE PARAMETERS

For each model, we detail the encryption parameters used in our experiments. Our framework automatically determines the RNS moduli according to the specified sizes $|q_i|$. We chose the smallest values of $\Delta$ and $|q_i|$ that preserve accuracy during private inference and adjust $N$ for $\approx$128-bit security. The moduli are ordered as $q_0, \ldots, q_L, P$, where $q_0$ holds the output, $q_1, \ldots, q_L$ are for rescaling, and $P$ is for modulus expansion in relinearization. FHE terminology is provided in §A.

**ResNet-18:** $N = 2^{15}$, $\Delta = 2^{22}$, $\log_2 Q = 18 \cdot 44 + 23 + 54 = 869$. Moduli: {0x100000020001, 0x100000050001, 0x100000090001, 0x1000000b0001, 0xfffffcf0001, 0x100000180001, 0x1000001a0001, 0xfffffc60001, 0x1000002c0001, 0xfffffb70001, 0x1000002d0001, 0x1000003c0001, 0xfffffb50001, 0x1000003e0001, 0xfffffaf0001, 0x100000480001, 0xffff-fac0001, 0x100000570001, 0x820001, 0x3ffffffd60001}.

**ResNet-20:** $N = 2^{15}$, $\Delta = 2^{21}$, $\log_2 Q = 20 \cdot 42 + 22 + 44 = 906$. Moduli: {0x400000b0001, 0x3ffffe80001, 0x400002f0001, 0x3ffffd20001, 0x40000330001, 0x3ffffca0001, 0x40000390001, 0x3ffffc30001, 0x400003b0001, 0x3ffffbe0001, 0x400004d0001, 0x3ffff850001, 0x40000560001, 0x400005c0001, 0x3ffff7b0001, 0x400006c0001, 0x3ffff550001, 0x40000770001, 0x3ffff4f0001, 0x400007a0001, 0x390001, 0x100000020001}.

**ResNet-32:** $N = 2^{16}$, $\Delta = 2^{26}$, $\log_2 Q = 32 \cdot 52 + 27 + 54 = 1745$. Moduli: {0x10000000060001, 0xffffffff00001, 0x10000000180001, 0xfffffffe40001, 0x10000000200001, 0xfffffffe20001, 0x100000003e0001, 0xfffffffbe0001, 0x10000000500001, 0xfffffffa60001, 0x100000006e0001, 0xffffffff820001, 0x100000007e0001, 0xffffffff480001, 0x10000000960001, 0xffffffff280001, 0x10000000c80001, 0x10000000d80001, 0xfffffffed60001, 0x10000000ec0001, 0xfffffffec40001, 0x10000000fc0001, 0xfffffffeb00001, 0x100000010e0001, 0xfffffffe9e0001, 0x10000001380001, 0xfffffffe9a0001, 0x100000016a0001, 0xfffffffe940001, 0x10000001bc0001, 0xfffffffe6a0001, 0x10000001be0001, 0x8020001, 0x3ffffffd60001}.

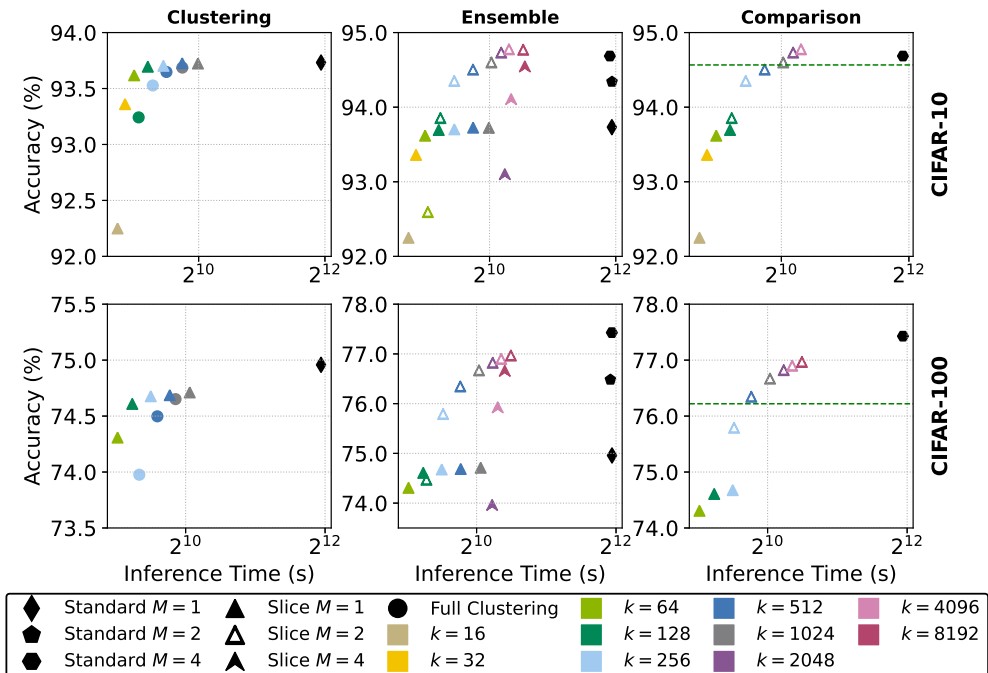

Figure 7: Accuracy vs. private inference time for ResNet-18 on CIFAR-10/100. Ensembles are evaluated with $M \in \{1, 2, 4\}$ ($M = 1$ is a single model). We compare Standard ensembles (no clustering), Slice ensembles (slice-wise clustering), and Full Clustering. Colored squares represent centroid counts ($k = 16, \ldots, 8192$), and the same color is used for the same centroid count in both slice and full clustering. The green dashed line shows plaintext ReLU accuracy.

## H.2 RESULTS ON RESNET18

Figure 7 shows the trade-offs between accuracy and inference time for ResNet-18 on CIFAR-10 and CIFAR-100. Results show similar traits to ResNet-20, leading to similar conclusions. In the clustering plots (left column), both *Full* and *Slice Clustering* reduce inference time compared to *Standard*. At small centroid counts ($k \leq 32$), accuracy drops noticeably due to a lack of representative centroids. *Slice Clustering* consistently achieves higher accuracy than *Full Clustering*, benefiting from its finer-grained parameter representation. When the centroid count is sufficiently large ($k \geq 64$), *Slice Clustering* reaches the accuracy of *Standard* while still reducing latency substantially.

The middle column plots compare *Standard* and *Slice Clustering* ensembles with $M \in \{1, 2, 4\}$. *Standard* gains accuracy as $M$ increases, while *Slice Clustering* shows more variable behavior because each centroid encodes an $M$-dimensional parameter space; e.g. *Slice-4* underperforms relative to *Slice-2* for small $k$. *Slice Clustering* provides a regularization benefit and, with larger centroid counts, *Slice-2* achieves higher accuracy than *Standard-2* and nearly matches *Standard-4*.

The right column presents the Pareto front of *Slice Clustering* and compares it to *Standard-4*. *Slice Clustering* shifts the curve toward lower latency while staying close to *Standard* in accuracy. On both datasets, *Slice Clustering* fully recovers plaintext ReLU accuracy and sometimes even surpasses it, demonstrating that accuracy can be preserved while reducing inference time.

## H.3 EXPERIMENTAL RESULTS

Tables 3, 4, and 5 report the accuracy, inference time, and peak memory usage of ResNet-18, ResNet-20, and ResNet-32, respectively, on the CIFAR-10 and CIFAR-100 datasets using the Standard, Full Clustering, and Slice Clustering methods. For Full Clustering and Slice Clustering, we varied the number of centroids as $k = 2^i \ \forall i \in \{1, \ldots, 10\}$. For Standard and Slice Clustering, we additionally evaluated ensemble models with $M \in 1, 2, 4$. In the case of Slice Clustering with $M \geq 2$, we extended the centroid range to $k = 2^i \ \forall i \in \{1, \ldots, 12\}$ in ResNet-18 and ResNet-20. We did not extend this configuration to ResNet-32 due to system memory limitations (§5.1).

Table 3: Accuracy, inference time, and peak memory for CIFAR-10 and CIFAR-100 on ResNet-18.

**Standard**

| M | Accuracy (%) μ | σ | Inf. Time (s) μ | σ | Memory (GB) |
|---|---|---|---|---|---|
| 1 | 93.73 | 0.27 | 3925 | 51 | 292 |
| 2 | 94.34 | 0.16 | 3930 | 54 | 314 |
| 4 | 94.69 | 0.12 | 3840 | 44 | 347 |

| M | Accuracy (%) μ | σ | Inf. Time (s) μ | σ | Memory (GB) |
|---|---|---|---|---|---|
| 1 | 74.96 | 0.51 | 3935 | 79 | 338 |
| 2 | 76.48 | 0.28 | 3874 | 40 | 355 |
| 4 | 77.43 | 0.24 | 3929 | 54 | 390 |

**Full Clustering (M = 1)**

| k | Accuracy (%) μ | σ | Inf. Time (s) μ | σ | Memory (GB) |
|---|---|---|---|---|---|
| 2 | 10.16 | 0.72 | 404 | 16 | 197 |
| 4 | 10.14 | 0.89 | 406 | 18 | 200 |
| 8 | 9.91 | 1.48 | 420 | 31 | 202 |
| 16 | 15.89 | 7.51 | 427 | 27 | 204 |
| 32 | 70.18 | 14.19 | 439 | 28 | 204 |
| 64 | 91.42 | 1.25 | 481 | 31 | 204 |
| 128 | 93.24 | 0.29 | 530 | 34 | 205 |
| 256 | 93.53 | 0.26 | 619 | 36 | 209 |
| 512 | 93.65 | 0.25 | 719 | 40 | 221 |
| 1024 | 93.69 | 0.28 | 856 | 34 | 245 |

| k | Accuracy (%) μ | σ | Inf. Time (s) μ | σ | Memory (GB) |
|---|---|---|---|---|---|
| 2 | 1.00 | 0.07 | 414 | 25 | 240 |
| 4 | 1.02 | 0.12 | 413 | 17 | 242 |
| 8 | 1.08 | 0.17 | 426 | 23 | 244 |
| 16 | 2.01 | 1.10 | 430 | 25 | 244 |
| 32 | 32.33 | 13.66 | 441 | 23 | 246 |
| 64 | 68.01 | 4.00 | 484 | 34 | 246 |
| 128 | 72.62 | 1.04 | 554 | 41 | 247 |
| 256 | 73.98 | 0.68 | 646 | 44 | 250 |
| 512 | 74.50 | 0.70 | 773 | 53 | 263 |
| 1024 | 74.65 | 0.69 | 926 | 58 | 286 |

**Slice Clustering (M = 1)**

| k | Accuracy (%) μ | σ | Inf. Time (s) μ | σ | Memory (GB) |
|---|---|---|---|---|---|
| 2 | 9.79 | 0.76 | 418 | 32 | 198 |
| 4 | 12.66 | 3.02 | 409 | 23 | 200 |
| 8 | 81.28 | 6.04 | 412 | 23 | 203 |
| 16 | 92.25 | 0.68 | 420 | 19 | 204 |
| 32 | 93.36 | 0.33 | 456 | 28 | 204 |
| 64 | 93.62 | 0.27 | 504 | 30 | 204 |
| 128 | 93.70 | 0.26 | 586 | 30 | 205 |
| 256 | 93.70 | 0.28 | 696 | 34 | 214 |
| 512 | 93.73 | 0.27 | 855 | 58 | 232 |
| 1024 | 93.72 | 0.27 | 1016 | 57 | 266 |

| k | Accuracy (%) μ | σ | Inf. Time (s) μ | σ | Memory (GB) |
|---|---|---|---|---|---|
| 2 | 1.07 | 0.17 | 426 | 16 | 239 |
| 4 | 2.76 | 1.17 | 413 | 13 | 242 |
| 8 | 42.21 | 13.71 | 420 | 24 | 244 |
| 16 | 69.28 | 2.37 | 443 | 25 | 246 |
| 32 | 72.85 | 0.92 | 466 | 27 | 245 |
| 64 | 74.31 | 0.77 | 520 | 31 | 246 |
| 128 | 74.61 | 0.73 | 602 | 36 | 248 |
| 256 | 74.68 | 0.71 | 724 | 39 | 256 |
| 512 | 74.69 | 0.73 | 874 | 49 | 274 |
| 1024 | 74.71 | 0.74 | 1066 | 66 | 308 |

**Slice Clustering (M = 2)**

| k | Accuracy (%) μ | σ | Inf. Time (s) μ | σ | Memory (GB) |
|---|---|---|---|---|---|
| 64 | 92.59 | 0.45 | 520 | 43 | 222 |
| 128 | 93.85 | 0.15 | 597 | 38 | 222 |
| 256 | 94.35 | 0.10 | 695 | 22 | 230 |
| 512 | 94.50 | 0.09 | 854 | 28 | 250 |
| 1024 | 94.60 | 0.08 | 1045 | 61 | 284 |
| 2048 | 94.73 | 0.07 | 1164 | 36 | 394 |
| 4096 | 94.78 | 0.04 | 1268 | 42 | 643 |
| 8192 | 94.77 | 0.05 | 1482 | 59 | 1148 |

| k | Accuracy (%) μ | σ | Inf. Time (s) μ | σ | Memory (GB) |
|---|---|---|---|---|---|
| 64 | 68.93 | 1.94 | 524 | 10 | 263 |
| 128 | 74.47 | 0.59 | 622 | 23 | 263 |
| 256 | 75.79 | 0.43 | 734 | 51 | 273 |
| 512 | 76.35 | 0.18 | 871 | 37 | 292 |
| 1024 | 76.67 | 0.12 | 1049 | 23 | 326 |
| 2048 | 76.82 | 0.14 | 1202 | 72 | 435 |
| 4096 | 76.90 | 0.08 | 1308 | 18 | 681 |
| 8192 | 76.97 | 0.07 | 1440 | 13 | 1174 |

**Slice Clustering (M = 4)**

| k | Accuracy (%) μ | σ | Inf. Time (s) μ | σ | Memory (GB) |
|---|---|---|---|---|---|
| 64 | 18.38 | 4.80 | 526 | 27 | 256 |
| 128 | 35.79 | 8.02 | 621 | 45 | 258 |
| 256 | 66.88 | 5.19 | 728 | 51 | 265 |
| 512 | 84.28 | 2.38 | 866 | 28 | 285 |
| 1024 | 91.47 | 0.46 | 1035 | 69 | 319 |
| 2048 | 93.10 | 0.21 | 1211 | 57 | 428 |
| 4096 | 94.11 | 0.14 | 1298 | 17 | 679 |
| 8192 | 94.55 | 0.11 | 1509 | 22 | 1183 |

| k | Accuracy (%) μ | σ | Inf. Time (s) μ | σ | Memory (GB) |
|---|---|---|---|---|---|
| 64 | 1.73 | 0.41 | 538 | 26 | 297 |
| 128 | 5.31 | 0.94 | 615 | 27 | 299 |
| 256 | 29.69 | 2.16 | 750 | 47 | 308 |
| 512 | 57.88 | 2.20 | 885 | 16 | 326 |
| 1024 | 68.98 | 1.13 | 1057 | 46 | 360 |
| 2048 | 73.96 | 0.38 | 1195 | 64 | 469 |
| 4096 | 75.93 | 0.22 | 1263 | 24 | 716 |
| 8192 | 76.66 | 0.12 | 1355 | 17 | 1208 |

Table 4: Accuracy, inference time, and peak memory for CIFAR-10 and CIFAR-100 on ResNet-20.

| | Standard | | | | | | Standard | | | | |
|---|---|---|---|---|---|---|---|---|---|---|---|
| M | Accuracy (%) | | Inf. Time (s) | | Memory | M | Accuracy (%) | | Inf. Time (s) | | Memory |
| | $\mu$ | $\sigma$ | $\mu$ | $\sigma$ | (GB) | | $\mu$ | $\sigma$ | $\mu$ | $\sigma$ | (GB) |
| 1 | 94.06 | 0.26 | 1886 | 30 | 279 | 1 | 73.89 | 0.37 | 1902 | 26 | 305 |
| 2 | 94.71 | 0.11 | 1884 | 41 | 300 | 2 | 76.59 | 0.25 | 1874 | 27 | 320 |
| 4 | 95.04 | 0.10 | 1886 | 32 | 330 | 4 | 77.97 | 0.13 | 1904 | 30 | 351 |

| | Full Clustering (M = 1) | | | | | | Full Clustering (M = 1) | | | | |
|---|---|---|---|---|---|---|---|---|---|---|---|
| k | Accuracy (%) | | Inf. Time (s) | | Memory | k | Accuracy (%) | | Inf. Time (s) | | Memory |
| | $\mu$ | $\sigma$ | $\mu$ | $\sigma$ | (GB) | | $\mu$ | $\sigma$ | $\mu$ | $\sigma$ | (GB) |
| 2 | 10.08 | 0.26 | 242 | 10 | 175 | 2 | 1.03 | 0.07 | 246 | 6 | 208 |
| 4 | 10.08 | 0.32 | 246 | 6 | 190 | 4 | 1.08 | 0.23 | 246 | 9 | 211 |
| 8 | 10.17 | 0.89 | 250 | 12 | 192 | 8 | 1.17 | 0.24 | 252 | 7 | 212 |
| 16 | 14.44 | 6.34 | 260 | 13 | 192 | 16 | 2.05 | 1.58 | 258 | 8 | 213 |
| 32 | 78.18 | 8.99 | 264 | 7 | 193 | 32 | 32.59 | 14.07 | 272 | 16 | 214 |
| 64 | 92.46 | 0.95 | 288 | 10 | 193 | 64 | 68.18 | 3.24 | 290 | 11 | 214 |
| 128 | 93.77 | 0.37 | 334 | 12 | 194 | 128 | 71.56 | 1.19 | 334 | 15 | 214 |
| 256 | 94.00 | 0.26 | 407 | 22 | 198 | 256 | 73.39 | 0.49 | 392 | 9 | 218 |
| 512 | 94.02 | 0.26 | 475 | 18 | 212 | 512 | 73.75 | 0.35 | 473 | 20 | 232 |
| 1024 | 94.05 | 0.26 | 553 | 25 | 248 | 1024 | 73.85 | 0.35 | 563 | 28 | 264 |

| | Slice Clustering (M = 1) | | | | | | Slice Clustering (M = 1) | | | | |
|---|---|---|---|---|---|---|---|---|---|---|---|
| k | Accuracy (%) | | Inf. Time (s) | | Memory | k | Accuracy (%) | | Inf. Time (s) | | Memory |
| | $\mu$ | $\sigma$ | $\mu$ | $\sigma$ | (GB) | | $\mu$ | $\sigma$ | $\mu$ | $\sigma$ | (GB) |
| 2 | 10.05 | 0.41 | 249 | 18 | 186 | 2 | 1.12 | 0.22 | 254 | 10 | 207 |
| 4 | 14.59 | 4.33 | 244 | 7 | 190 | 4 | 2.11 | 0.62 | 255 | 13 | 210 |
| 8 | 85.17 | 2.66 | 247 | 7 | 192 | 8 | 49.94 | 5.99 | 264 | 21 | 212 |
| 16 | 93.20 | 0.45 | 262 | 6 | 193 | 16 | 69.60 | 1.71 | 262 | 7 | 214 |
| 32 | 93.87 | 0.32 | 276 | 9 | 194 | 32 | 72.98 | 0.45 | 292 | 12 | 214 |
| 64 | 94.01 | 0.28 | 314 | 9 | 194 | 64 | 73.68 | 0.38 | 323 | 13 | 214 |
| 128 | 94.01 | 0.28 | 366 | 14 | 194 | 128 | 73.85 | 0.38 | 377 | 18 | 215 |
| 256 | 94.04 | 0.27 | 442 | 10 | 205 | 256 | 73.89 | 0.34 | 452 | 17 | 226 |
| 512 | 94.05 | 0.26 | 536 | 26 | 226 | 512 | 73.89 | 0.36 | 538 | 10 | 247 |
| 1024 | 94.05 | 0.26 | 612 | 19 | 288 | 1024 | 73.88 | 0.36 | 626 | 21 | 309 |

| | Slice Clustering (M = 2) | | | | | | Slice Clustering (M = 2) | | | | |
|---|---|---|---|---|---|---|---|---|---|---|---|
| k | Accuracy (%) | | Inf. Time (s) | | Memory | k | Accuracy (%) | | Inf. Time (s) | | Memory |
| | $\mu$ | $\sigma$ | $\mu$ | $\sigma$ | (GB) | | $\mu$ | $\sigma$ | $\mu$ | $\sigma$ | (GB) |
| 64 | 92.52 | 0.56 | 312 | 9 | 209 | 64 | 68.98 | 0.97 | 321 | 14 | 230 |
| 128 | 94.34 | 0.10 | 370 | 8 | 211 | 128 | 74.31 | 0.44 | 380 | 16 | 230 |
| 256 | 94.81 | 0.08 | 443 | 20 | 219 | 256 | 76.08 | 0.25 | 464 | 7 | 241 |
| 512 | 94.93 | 0.08 | 531 | 31 | 242 | 512 | 76.65 | 0.19 | 541 | 22 | 263 |
| 1024 | 95.08 | 0.07 | 611 | 14 | 304 | 1024 | 76.99 | 0.11 | 614 | 22 | 324 |
| 2048 | 95.11 | 0.06 | 679 | 7 | 458 | 2048 | 77.16 | 0.13 | 690 | 23 | 476 |
| 4096 | 95.08 | 0.04 | 814 | 60 | 768 | 4096 | 77.12 | 0.12 | 746 | 17 | 781 |
| 8192 | 95.08 | 0.03 | 885 | 37 | 1389 | 8192 | 77.23 | 0.09 | 897 | 26 | 1390 |

| | Slice Clustering (M = 4) | | | | | | Slice Clustering (M = 4) | | | | |
|---|---|---|---|---|---|---|---|---|---|---|---|
| k | Accuracy (%) | | Inf. Time (s) | | Memory | k | Accuracy (%) | | Inf. Time (s) | | Memory |
| | $\mu$ | $\sigma$ | $\mu$ | $\sigma$ | (GB) | | $\mu$ | $\sigma$ | $\mu$ | $\sigma$ | (GB) |
| 64 | 12.79 | 2.88 | 332 | 14 | 240 | 64 | 1.72 | 0.48 | 341 | 13 | 261 |
| 128 | 25.95 | 4.81 | 389 | 7 | 242 | 128 | 3.86 | 0.96 | 407 | 10 | 262 |
| 256 | 57.23 | 3.93 | 455 | 21 | 251 | 256 | 19.97 | 2.42 | 463 | 19 | 271 |
| 512 | 83.18 | 1.59 | 534 | 10 | 274 | 512 | 52.26 | 2.81 | 546 | 8 | 294 |
| 1024 | 91.45 | 0.46 | 603 | 18 | 335 | 1024 | 68.37 | 0.63 | 631 | 14 | 356 |
| 2048 | 93.89 | 0.26 | 648 | 9 | 489 | 2048 | 73.73 | 0.53 | 673 | 11 | 508 |
| 4096 | 94.65 | 0.16 | 722 | 12 | 799 | 4096 | 76.44 | 0.31 | 765 | 9 | 813 |
| 8192 | 94.90 | 0.10 | 815 | 17 | 1421 | 8192 | 77.62 | 0.17 | 855 | 16 | 1421 |

Table 5: Accuracy, inference time, and peak memory for CIFAR-10 and CIFAR-100 on ResNet-32.

### Standard (CIFAR-10)

| M | Accuracy (%) $\mu$ | $\sigma$ | Inf. Time (s) $\mu$ | $\sigma$ | Memory (GB) |
|---|---|---|---|---|---|
| 1 | 93.86 | 0.24 | 9405 | 109 | 1332 |
| 2 | 94.69 | 0.18 | 8854 | 80 | 1394 |
| 4 | 94.99 | 0.13 | 8813 | 63 | 1441 |

### Standard (CIFAR-100)

| M | Accuracy (%) $\mu$ | $\sigma$ | Inf. Time (s) $\mu$ | $\sigma$ | Memory (GB) |
|---|---|---|---|---|---|
| 1 | 73.93 | 0.53 | 8865 | 132 | 1401 |
| 2 | 76.79 | 0.34 | 8784 | 60 | 1429 |
| 4 | 78.34 | 0.23 | 8776 | 9 | 1438 |

### Full Clustering (M = 1) (CIFAR-10)

| k | Accuracy (%) $\mu$ | $\sigma$ | Inf. Time (s) $\mu$ | $\sigma$ | Memory (GB) |
|---|---|---|---|---|---|
| 2 | 10.12 | 0.22 | 1578 | 63 | 842 |
| 4 | 10.13 | 0.21 | 1538 | 3 | 867 |
| 8 | 10.12 | 0.22 | 1588 | 79 | 879 |
| 16 | 10.35 | 1.35 | 1570 | 8 | 893 |
| 32 | 37.13 | 16.87 | 1578 | 3 | 898 |
| 64 | 86.61 | 3.69 | 1703 | 80 | 901 |
| 128 | 92.82 | 0.47 | 2131 | 27 | 902 |
| 256 | 93.70 | 0.32 | 2164 | 151 | 916 |
| 512 | 93.84 | 0.27 | 2495 | 164 | 987 |
| 1024 | 93.86 | 0.26 | 2898 | 174 | 1155 |

### Full Clustering (M = 1) (CIFAR-100)

| k | Accuracy (%) $\mu$ | $\sigma$ | Inf. Time (s) $\mu$ | $\sigma$ | Memory (GB) |
|---|---|---|---|---|---|
| 2 | 0.98 | 0.07 | 1544 | 5 | 889 |
| 4 | 0.98 | 0.07 | 1577 | 20 | 908 |
| 8 | 0.97 | 0.07 | 1540 | 24 | 914 |
| 16 | 1.03 | 0.21 | 1547 | 27 | 929 |
| 32 | 3.91 | 2.63 | 1565 | 13 | 942 |
| 64 | 52.92 | 7.49 | 1686 | 44 | 945 |
| 128 | 70.26 | 1.24 | 1819 | 5 | 947 |
| 256 | 73.18 | 0.38 | 2013 | 33 | 954 |
| 512 | 73.75 | 0.37 | 2381 | 70 | 1020 |
| 1024 | 73.89 | 0.33 | 2822 | 53 | 1159 |

### Slice Clustering (M = 1) (CIFAR-10)

| k | Accuracy (%) $\mu$ | $\sigma$ | Inf. Time (s) $\mu$ | $\sigma$ | Memory (GB) |
|---|---|---|---|---|---|
| 2 | 10.12 | 1.07 | 1605 | 57 | 857 |
| 4 | 10.00 | 0.99 | 1573 | 24 | 879 |
| 8 | 34.61 | 12.19 | 1601 | 65 | 893 |
| 16 | 89.08 | 2.44 | 1559 | 6 | 901 |
| 32 | 93.12 | 0.49 | 1761 | 90 | 902 |
| 64 | 93.75 | 0.29 | 1833 | 54 | 905 |
| 128 | 93.84 | 0.26 | 2077 | 111 | 905 |
| 256 | 93.86 | 0.27 | 2488 | 185 | 950 |
| 512 | 93.85 | 0.25 | 2849 | 128 | 1060 |
| 1024 | 93.86 | 0.24 | 3317 | 192 | 1373 |

### Slice Clustering (M = 1) (CIFAR-100)

| k | Accuracy (%) $\mu$ | $\sigma$ | Inf. Time (s) $\mu$ | $\sigma$ | Memory (GB) |
|---|---|---|---|---|---|
| 2 | 0.99 | 0.08 | 1559 | 11 | 894 |
| 4 | 1.00 | 0.09 | 1526 | 27 | 920 |
| 8 | 9.67 | 5.64 | 1586 | 11 | 936 |
| 16 | 60.50 | 4.61 | 1576 | 8 | 942 |
| 32 | 71.75 | 0.80 | 1679 | 19 | 946 |
| 64 | 73.49 | 0.36 | 1803 | 9 | 947 |
| 128 | 73.83 | 0.33 | 2027 | 30 | 946 |
| 256 | 73.91 | 0.33 | 2379 | 83 | 993 |
| 512 | 73.93 | 0.33 | 2815 | 60 | 1104 |
| 1024 | 73.94 | 0.53 | 3313 | 57 | 1417 |

### Slice Clustering (M = 2) (CIFAR-10)

| k | Accuracy (%) $\mu$ | $\sigma$ | Inf. Time (s) $\mu$ | $\sigma$ | Memory (GB) |
|---|---|---|---|---|---|
| 64 | 85.12 | 4.61 | 1831 | 69 | 927 |
| 128 | 92.75 | 0.58 | 2069 | 79 | 933 |
| 256 | 94.06 | 0.21 | 2388 | 156 | 977 |
| 512 | 94.59 | 0.12 | 2788 | 167 | 1087 |
| 1024 | 94.76 | 0.12 | 3093 | 11 | 1399 |

### Slice Clustering (M = 2) (CIFAR-100)

| k | Accuracy (%) $\mu$ | $\sigma$ | Inf. Time (s) $\mu$ | $\sigma$ | Memory (GB) |
|---|---|---|---|---|---|
| 64 | 52.29 | 6.60 | 1791 | 17 | 968 |
| 128 | 70.20 | 1.26 | 2014 | 44 | 972 |
| 256 | 74.51 | 0.42 | 2340 | 52 | 1019 |
| 512 | 76.30 | 0.25 | 2809 | 38 | 1128 |
| 1024 | 76.77 | 0.24 | 3106 | 1 | 1437 |

### Slice Clustering (M = 4) (CIFAR-10)

| k | Accuracy (%) $\mu$ | $\sigma$ | Inf. Time (s) $\mu$ | $\sigma$ | Memory (GB) |
|---|---|---|---|---|---|
| 64 | 9.27 | 0.65 | 1845 | 89 | 979 |
| 128 | 14.20 | 2.55 | 2096 | 100 | 983 |
| 256 | 30.01 | 9.08 | 2409 | 149 | 1030 |
| 512 | 69.18 | 5.72 | 2842 | 193 | 1136 |
| 1024 | 88.24 | 1.02 | 3379 | 64 | 1450 |

### Slice Clustering (M = 4) (CIFAR-100)

| k | Accuracy (%) $\mu$ | $\sigma$ | Inf. Time (s) $\mu$ | $\sigma$ | Memory (GB) |
|---|---|---|---|---|---|
| 64 | 0.92 | 0.00 | 1817 | 22 | 1019 |
| 128 | 0.92 | 0.01 | 2152 | 163 | 1007 |
| 256 | 1.79 | 0.74 | 2614 | 126 | 1030 |
| 512 | 16.68 | 6.66 | 2829 | 291 | 1206 |
| 1024 | 47.96 | 5.14 | 3408 | 48 | 1452 |

