# OpenReview forum: "PAPER: Privacy-Preserving ResNet Models using Low-Degree Polynomial Approximations and Structural Optimizations on Leveled FHE"
_ICLR.cc/2026/Conference — Submitted to ICLR 2026_

### Official Review · Reviewer_xdM9 · 2025-10-31

**Soundness:** 3
**Presentation:** 3
**Contribution:** 3
**Rating:** 6
**Confidence:** 3

**Summary:**

This paper proposes a new method for privacy-preserving inference of deep convolutional neural networks under Fully Homomorphic Encryption (FHE). The authors focus on the ResNet architecture and present the first method that enables large models like ResNet-18, ResNet-20, and ResNet-32 to run entirely under Leveled FHE (LFHE) without bootstrapping. Their key idea is to use a degree-2 polynomial approximation of ReLU combined with a penalty-based training strategy to stabilize training. Furthermore, they introduce three structural optimization techniques (node fusing, weight redistribution, and tower reuse) to reduce the multiplicative depth, as well as co-design techniques like parameter clustering and ensemble inference that utilize unused ciphertext slots. Experiments on CIFAR-10 and CIFAR-100 show that their accuracy is comparable to plaintext ReLU models, and their inference speed is up to 4x faster than existing FHE-based methods (like AutoFHE and MPCNN).

**Strengths:**

1. It is the first work to demonstrate deep ResNet models (ResNet-18/20/32) running on Leveled FHE *without* requiring any bootstrapping. This is a highly significant result, as bootstrapping is the single largest performance bottleneck in the field. The authors achieve this by systematically attacking the total multiplicative depth, starting with a novel training strategy (§2) that successfully stabilizes a degree-2 polynomial activation—achieving the theoretical minimum depth for that component.
2. The minimal-depth activation is complemented by a suite of three novel structural optimizations (§3). The "node fusing" and "weight redistribution" are clever techniques for collapsing linear operations, but the "Tower Reuse" concept is particularly insightful, as it cleverly manipulates the CKKS scheme's parameters to reduce FHE levels even further.
3. The co-design strategy presented in §4 is highly innovative. The idea of exploiting "unused slots" in the HW layout to pack an ensemble of models is an excellent example of turning a layout inefficiency into a powerful feature. It provides a near "zero-cost" mechanism (in terms of inference time) to claw back the accuracy lost from the low-degree polynomial approximation. The subsequent introduction of "slice ensemble clustering" to mitigate the new memory/encoding overhead created by this ensemble demonstrates a deep, holistic understanding of the FHE-ML trade-offs.
4. All these claims are well-supported by both solid theoretical derivations and credible experimental validation.

**Weaknesses:**

Please refer to my questions below.

**Questions:**

Comment 1: Query regarding the Numerical Stability of Weight Redistribution (Sec. 3.2) under Fixed-Point Precision Constraints

The "Weight Redistribution" introduced in Section 3.2 is a very clever structural optimization, aiming to reduce the multiplicative depth of operations like polynomial activations to their theoretical minimum. This is significant for reducing the required FHE levels $L$.

A core claim of the paper is that this technique can be applied "while maintaining functional equivalence." Appendix E.2 provides detailed mathematical derivations proving this algebraic equivalence in the ideal domain of real numbers ($\\mathbb{R}$).

However, my question concerns how this ideal algebraic equivalence holds up when applied within the finite fixed-point precision framework set by the paper (i.e., $b$-bit precision as described in Sec. 2.2 and specified as $b=10$ in Sec. 5.1).

Specifically, in the "Update Forward" $\\rightarrow$ "Receivers" part (Sec. E.2.1), the coefficient update rule for a polynomial receiver is given by $\\bar{c}\i = c\i \\upsilon^i$, where $\\upsilon$ is a coefficient from the "donor" layer (e.g., $\\upsilon = c\d$).

I have the following questions:

1.  Regarding Exponential Coefficient Scaling: The update rule $\\bar{c}\i = c\i \\upsilon^i$ is exponential in nature. Could the authors clarify the impact of this? For example, if a "donor" has $\\upsilon = 1.5$ (a very reasonable value), the $c\2$ term of a 2nd-degree polynomial "receiver" will become $\\bar{c}\2 = c\2 \\cdot (1.5)^2 = 2.25 \\cdot c\2$, an amplification of 2.25x.

2.  Potential Conflict with Fixed-Point Precision: As shown in the example, in a deep network like ResNet, this redistribution operation may occur multiple times in sequence. If this operation, with $\\upsilon=1.5$, occurs just twice in a row, the same coefficient would grow by $2.25^2 \\approx 5.06$ times. Does this exponential amplification of coefficients (or exponential shrinking if $\\upsilon < 1.0$) not risk pushing coefficients to rapidly exceed the representable range of $b=10$ (causing Overflow), or fall below the precision of $b=10$ (causing Underflow or truncation to zero)?

3.  Guarantee of "Functional Equivalence": Given this potential risk of overflow or precision loss, have the authors conducted numerical range analyses or experiments to ensure that all redistributed coefficients ($\\bar{c}\i$ and $\\bar{w}\i$) remain strictly within the $b=10$ fixed-point precision range in all cases (or in the post-training models)?


Could the authors please clarify how "functional equivalence" is rigorously maintained in the practical, fixed-point FHE environment?

Comment 2: Query on the Generality of the "Zero-Overhead" Assumption for Ensemble Inference in Sec. 4.3

The authors propose a core co-design technique in Sec. 4.1 and 4.3: leveraging "unused slots" in the HW layout to pack $M$ ensemble models, thereby achieving ensemble inference with "no extra computation or memory overhead." This is critical for recovering the accuracy lost to polynomial approximation.

My query concerns the generality of this "zero-overhead" assumption.

1.  Dependence on Precondition: This technique appears to be entirely dependent on the precondition that the input data's spatial dimensions ($h \\times w$) are significantly smaller than the FHE ciphertext's slot capacity (typically $N/2$ in CKKS).

2.  Evidence in Paper: In the paper's experiments (Sec. 5.1, H.1), $N = 2^{15}$ (i.e., $N/2 = 16,384$ available slots), while the CIFAR-10 input is $h \\times w = 32 \\times 32 = 1024$. Clearly, $1024 \\ll 16,384$, which leaves 94% of "unused slots" available for packing.

3.  Limitation on Generality: The authors present this as a "co-design technique" in Sec. 4. However, for many non-"toy" computer vision tasks, it is common for the input size $h \\times w$ to approach or even exceed the slot capacity $N/2$. For example, on ImageNet (a standard benchmark), the input is $h \\times w = 224 \\times 224 = 50,176$. This size ($50,176$) is far larger than the $16,384$ available slots. In this scenario, not only are there no "unused slots" for packing, but even a single input channel cannot fit into one ciphertext. Does this mean the core advantage of this "zero-cost" ensemble fails completely in this standard scenario?


Could the authors please clarify the boundary conditions for this key technique more explicitly in the paper? Specifically, to acknowledge that its "zero-overhead" nature is highly contingent on the $h \\times w \\ll N/2$ condition.

Comment 3: Query on the Stability and Reproducibility of the Training Strategy (Sec 2.2)

A key achievement of this paper is the ability to train deep ResNets with a simple degree-2 polynomial, achieving an accuracy drop of only 1.3% (Sec 5.2). This result relies heavily on the "Activation Regularization" strategy (Sec 2.2) to prevent "escaping activations."

My query is regarding the stability and reproducibility of this training strategy.

1.  A "Three-Piece" Strategy: At the end of Sec 2.2, the authors state that the main $\\zeta$-penalty (Eq. 1) alone may still lead to instability. They then introduce two "additional strategies": (i) clipping pre-activations during training and (ii) introducing a warm-up schedule for $\\zeta$. This suggests that the final stable result depends on a "three-piece combination" of these components.

2.  Lack of Ablation: This combination may be sensitive to tuning and potentially difficult to reproduce. However, the paper does not seem to provide an ablation study that isolates the importance of the two "additional strategies."


To help assess the robustness of this method and the ease of its reproducibility, could the authors please clarify:

   Are all three components (the $\\zeta$-penalty, the hard-clipping, and the $\\zeta$-warm-up) necessary to achieve the reported stability and accuracy?
   What happens to the training stability and final accuracy if the "additional" hard-clipping strategy is disabled?
   Similarly, what happens if the $\\zeta$-warm-up schedule is disabled and the full penalty $\\zeta = 0.001$ is used from the start?

Understanding the necessity of each component is crucial for evaluating the overall robustness of this novel training approach.

Comment 4: Query on the Storage Cost of "Slice Ensemble Clustering" (Sec 4.2 & 4.3)

My query concerns the storage overhead of the proposed clustering techniques, particularly the 'Slice Ensemble Clustering' that is necessary for the final high-accuracy results.

1.  The Initial Claim: In Sec. 4.2, when "Slice Clustering" (for a single model) is introduced, the paper states that the "additional storage from slice-specific codebooks is modest." This is a key claim.

2.  The Cost Multiplier: However, in Sec. 4.3, this method is extended to "Slice Ensemble Clustering" (SEC) to support $M$ models. As described, this changes each codebook entry from a scalar to an $M$-dimensional vector. This logically implies that the total storage cost for all codebooks is multiplied by a factor of $M$ (e.g., $M=2$ or $M=4$).

3.  The Contradiction in Appendix H.3: This multiplied cost seems to be far from "modest." My concern is strongly supported by the authors' own statement in Appendix H.3:

    > "We did not extend this configuration \[Slice Clustering with M ≥ 2\] to ResNet-32 due to system memory limitations."

4.  The Question: This suggests that the actual storage cost of the final, high-accuracy "Slice Ensemble Clustering" technique is not "modest" at all, but is, in fact, a critical bottleneck that prevents the method from scaling even from ResNet-20 to ResNet-32.


Could the authors please clarify this apparent contradiction? How should one reconcile the claim of "modest" overhead in Sec. 4.2 with the "system memory limitations" encountered for ResNet-32 in Appendix H.3?

---

> ### Author Response · Authors · 2025-12-02
>
> **Answer to Comment 1:**
> The model equivalence only holds in the real numbers. When moving to 64-bit arithmetic, the pre and post weight redistribution models are no longer mathematically identical, but the numerical deviation is negligible. In our experiments we observed no measurable difference in the logits of the plaintext model before and after redistribution.
>
> In practice, the dominant sources of error are the Gaussian noise added to ciphertexts during encryption and the approximation error incurred through rescaling. Weight redistribution contributes no observable error.
>
> Weight redistribution is applied prior to encoding, so it cannot lead to overflow in the encrypted domain. The b=10 setting is unrelated to these operations. It is used only when defining the coefficients of the polynomial activation that approximates ReLU on [−c,c] before training. The scaling factor used during FHE computation is at least 2^{20}, which provides substantially higher representational precision than the polynomial fitting precision.
>
> **Answer to Comment 2:**
> There is no overhead as long as all model instances fit in a single ciphertext. Once additional ciphertexts are required, inference slows down. This limitation is inherent to all data layouts, since larger inputs always demand more ciphertexts. Our approach therefore slows down for larger inputs, but competing methods incur similar slowdowns under the same conditions. The slowdown is stepwise linear in all cases. We will discuss this condition in the revised manuscript.
>
> **Answer to Comment 3:**
> Clipping and warm-up are both needed in practice because the quadratic activation makes early-epoch dynamics especially unstable. With degree-2 polynomials the network has no natural damping outside the approximation interval, so pre-activations can quickly explode before the penalty term meaningfully constrains them. If clipping is removed the model often diverges in the first few epochs. If warm-up is removed and the full penalty is applied from the start the optimization becomes dominated by the penalty term and training stalls or converges to a poor solution.
> This behavior aligns with what PILLAR already established for higher-degree polynomials: activation clipping and gradual introduction of the regularization weight are essential for stabilizing training under fixed-point polynomial activations. Since these mechanisms were already shown to be necessary in prior work we did not repeat the ablation. In our setting they serve the same purpose and we did not observe stable convergence of deep ResNets with degree-2 activations when either component was removed. We will clarify this dependency in the revised manuscript.
>
> **Answer to Comment 4:**
> Codebooks are only used during training. Once centroids are defined, each weight is replaced by its representative centroid. There is no codebook in the final model, so codebooks do not affect inference memory, only training memory. Training is done in plaintext, so its memory cost is similar to standard training.
> The extra memory associated with Slice Clustering with $M=2$ or $M=4$ does not contradict the claim in Sec. 4.2. In Tables 3–5, the peak memory of Full Clustering, which uses a single codebook, and Slice Clustering, which uses a codebook per slice, is similar. This shows that the number of codebooks does not drive memory usage. Memory consumption is dominated by encoded weights in convolution layers. The maximum number of encoded plaintexts at any moment is K, because up to K unique weights exist, and this determines the peak memory.
> ResNet-32 uses a larger polynomial degree, which produces larger plaintexts and ciphertexts. When K is large, memory usage becomes very high. This limitation affects both Slice and Full Clustering. If Full Clustering were run on ResNet-32 with K=8192, it would also run out of memory. The bottleneck comes from plaintext size and the number of unique encodings, not from maintaining multiple codebooks.
>
> We will clarify this point in the manuscript to avoid any misunderstanding.

---

### Official Review · Reviewer_fn4b · 2025-11-01

**Soundness:** 3
**Presentation:** 3
**Contribution:** 2
**Rating:** 4
**Confidence:** 3

**Summary:**

This paper addresses the slow inference of HE-based CNNs, which traditionally rely on costly bootstrapping due to high-degree polynomial approximations of ReLU. This work is the first to execute deep CNNs like ResNet-18, -20, and -32 entirely under leveled FHE (LFHE) without bootstrapping. To achieve this, the authors propose a quadratic (degree-2) polynomial approximation of ReLU stabilized by a novel penalty-based training strategy. This is combined with structural optimizations—node fusing, weight redistribution, and tower reuse—to minimize the required multiplicative depth. Furthermore, parameter clustering and an ensemble technique are jointly applied to accelerate inference and recover accuracy. Experiments on CIFAR-10/100 demonstrate that this approach achieves up to 4x faster inference while matching the accuracy of plaintext ReLU models.

**Strengths:**

- The work addresses and resolves the fundamental challenges of polynomial approximation in FHE, namely coefficient truncation and escaping activations . The proposed **regularization with a clip-range penalty**, in particular, appears to be a generalizable technique applicable to the approximation of various functions in HE-based PPML, not just ReLU.
- The **Tower Reuse** and **Parameter Clustering**  techniques seem broadly applicable beyond this specific paper and could aid in reducing levels and complexity in other PPML models.
- It successfully implements relatively deep models (ResNet-20/32) using LFHE, achieving competitive performance in terms of both **accuracy and inference time**.

**Weaknesses:**

- The paper presents low-degree polynomial approximations and structural optimizations as its primary contributions. However, the polynomial approximation method itself directly adopts the 'quantization-aware polynomial fitting' from PILLAR (Diaa et al., 2024). Furthermore, the **training strategy** shares similarities with PILLAR, as both pursue the same objective (stabilizing activations) and utilize a clip function during training. Additionally, idea of introducing a regularization term to constrain the range of intermediate values—while effective—is a well-established concept, particularly in fields like model quantization (e.g., [1]), which learns clipping parameters via regularization). This existing body of work weakens the novelty claim of the training strategy.
- The "novel structural optimizations" of **Node Fusing** and **Weight Redistribution** might be considered minor or standard techniques in FHE-based PPML. For example, prior works like MPCNN have presented the fusing of Convolution and Batch Normalization (ConvBN), and AutoFHE also presents techniques for fusing consecutive operations (such as two convolutions) into a single one, treating them as implementation details rather than major, standalone contributions. Thus, these optimizations could be viewed more as clear specifications of implementation rather than truly novel techniques.

[1] Choi, Jungwook, et al. "Pact: Parameterized clipping activation for quantized neural networks." *arXiv preprint arXiv:1805.06085* (2018).

**Questions:**

- The "Tower Reuse" technique, which introduces sublevels to decouple the RNS modulus size from the scaling factor, bears a strong conceptual resemblance to prior work, notably **"**Grafting: Decoupled Scale Factors and Modulus in RNS-CKKS" [2]. What are the specific technical advantages or novel contributions that differentiate the proposed Tower Reuse method from the established Grafting technique?
- The paper posits that pre-encoding all weight combinations leads to "prohibitive memory usage" and "on-demand" encoding causes "heavy computational overhead". This is a critical premise for the utility of parameter clustering. However, is this overhead truly prohibitive? Is it not feasible to store pre-encoded plaintexts organized by their required level and layout? The paper states this is a common issue, yet the claim of "heavy" overhead lacks external citation (e.g., from a survey paper) to quantify this bottleneck.

[2] Cheon, Jung Hee, et al. "Grafting: Decoupled Scale Factors and Modulus in RNS-CKKS." *Cryptology ePrint Archive* (2024).

---

> ### Author Response · Authors · 2025-12-02
>
> **Advances Over Prior Work in Polynomial Activations and Depth Reduction.** We agree that prior work, including PILLAR, uses quantization-aware polynomial fitting and employs clipping during training. Our contribution is not the reuse of these ideas in isolation but the demonstration that they enable stable training of large CNNs with degree-2 activations, which PILLAR does not achieve. Our method reaches ReLU-level accuracy while using the theoretical minimum multiplicative depth, whereas PILLAR requires degree 4 and still incurs a larger accuracy drop. The paper also introduces structural optimizations that significantly reduce multiplicative depth in ResNet models, making leveled FHE inference possible without bootstrapping for the first time. These results show that, although we build part of our training on known components, their combination and practical effectiveness are novel and lead to measurable improvements in both accuracy and efficiency. This combination creates the first practical pipeline that simultaneously achieves ReLU-level accuracy, minimal multiplicative depth, and full LFHE execution of large ResNets, establishing a capability that prior work had not demonstrated.
>
> **Advances in Layer Fusion Weight Redistribution.** Conv-BN folding is indeed known and originates from prior work, as our paper itself cites. It is a standard graph-level simplification and not tied to FHE. However, the paper introduces three additional fusing patterns that are new: fusion of BN with polynomial activations, fusion of two BN-normalized residual branches into a single bivariate polynomial (polyskip), and fusion of BN with identity-skip connections. These forms do not appear in prior PPML work such as MPCNN or AutoFHE, which only fuse linear layers and BN. Weight redistribution is also novel. Prior PPML systems treat constant multipliers in pooling, BN, and polynomial activations as fixed costs, whereas this paper shows how to systematically normalize these operations and propagate compensating scalings across the computation graph to reduce multiplicative depth without changing network semantics. This mechanism is not present in MPCNN, AutoFHE, or other FHE-based PPML frameworks.
>
> **Answer to Questions:** Grafting decouples the scaling factor from the RNS moduli by assigning a different scaling factor to each level and using fractional rescaling, which introduces approximation error similar to standard rescaling. Tower reuse instead partitions each level into sublevels, allowing several multiplications within a single level, while requiring only one rescale operation after all sublevels have been used. Thus, it reduces the total number of rescale operations. Both approaches reduce the number of RNS towers, which speeds computation and decreases ciphertext size. Additionally, tower reuse reduces the number of rescaling operations, providing extra speedup and reducing the accumulated noise from rescaling. Grafting reports up to ~2× speedup, while tower reuse achieves ~3.6× speedup for ResNet-18 and ResNet-20 and ~3.8× speedup for ResNet-32.

---

### Official Review · Reviewer_P459 · 2025-11-01

**Soundness:** 3
**Presentation:** 3
**Contribution:** 2
**Rating:** 4
**Confidence:** 4

**Summary:**

This paper presents a series of technical optimizations for privacy-preserving ResNet inference under leveled FHE, successfully eliminating bootstrapping and achieving up to 4× speedups on CIFAR-10/100 datasets. While the individual techniques---including low-degree polynomial activations, node fusing, weight redistribution, and parameter clustering---are well-executed and mathematically sound, the work suffers from significant limitations that undermine its broader impact. The evaluation is restricted to outdated CNN architectures (ResNet-18/20/32) on toy datasets, with no discussion of whether these methods generalize to modern architectures like Transformers or Vision Transformers, which dominate current research. More critically, the lack of bootstrapping is not a feature but rather a symptom: it indicates the approach fundamentally relies on bounded multiplicative depth, which severely constrains model scale and type, making it unsuitable for contemporary deep learning. The paper essentially optimizes within a narrow, legacy setting without addressing whether the proposed techniques can be repurposed for the increasingly diverse and complex models that the PPML community actually needs to deploy. Unless the authors provide compelling evidence that these methods extend meaningfully to Transformers or other modern architectures, or propose a fundamentally new framework rather than incremental optimization of ResNets, the work remains a well-polished but ultimately narrow contribution to a shifting landscape.

**Strengths:**

This paper is technically solid and well-executed, successfully combining multiple complementary optimization strategies (low-degree polynomials, structural fusing, weight redistribution, and clever clustering) to achieve impressive quantitative gains—eliminating bootstrapping, reducing RNS levels by ~5×, and delivering 4× speedups on ResNets without sacrificing accuracy. The theoretical contributions are rigorous with proper mathematical proofs, and the ablation studies are thorough, making this a valuable reference for anyone working on FHE-based neural network inference within the specific constraints of leveled homomorphic encryption.

**Weaknesses:**

The fundamental limitation is that the absence of bootstrapping is not an achievement but rather a constraint—the method is fundamentally bounded by fixed multiplicative depth, which severely restricts the scale and type of models that can be supported, and there is absolutely no evidence or discussion that these techniques generalize beyond ResNets to modern architectures like Transformers or ViTs. The experimental evaluation is confined to outdated CNN architectures on toy datasets (CIFAR-10/100), which represent a narrow slice of what contemporary deep learning actually looks like, and the authors do not even acknowledge this limitation as a concern. The paper essentially performs impressive local optimization within a legacy setting without proposing any principled path forward for the diverse, evolving landscape of modern models—making it a well-crafted but ultimately myopic contribution to a field that has already moved well beyond ResNets.

**Questions:**

Can this approach actually scale to ImageNet-level datasets with the current setup?

---

> ### Author Response · Authors · 2025-12-02
>
> **Clarification on Bootstrapping.** Our work reduces the multiplicative depth of CNNs by nearly a factor of five and enables ResNet-18, ResNet-20, and ResNet-32 to run fully under leveled FHE without bootstrapping. The absence of bootstrapping for these models is not a limitation but a result of our depth-reduction techniques, which significantly expand the numbers of layers that can be executed before bootstrapping becomes necessary. Our techniques are also applicable to deeper models, reducing the number of bootstrapping layers required from several dozen to only a few; thus, improving inference time.
>
> **Clarifying our Focus on CNNs.** The method is designed for CNNs and is compatible with standard architectures such as AlexNet, VGG, and ResNets. We evaluate ResNets because they are the established benchmark in FHE-based private inference literature and represent a more challenging setting. Testing every CNN architecture is impractical due to the scale of experiments required, and concentrating on ResNets allows direct comparison with prior work. Research on private inference for Transformers is still at an early stage, while CNN-based private inference remains an active and unsolved problem in the FHE community. Current state-of-the-art methods for CNNs still require expensive bootstrapping to maintain accuracy. Our contribution addresses this challenge by making large CNNs practical under pure leveled FHE, demonstrating a capability previously thought unattainable.
>
> **Feasibility of Scaling to ImageNet and Answer to Q1**. Our choice of CIFAR-10 and CIFAR-100 follows established practice in non-interactive PPML, where ResNet-18, ResNet-20, and ResNet-32 are the standard benchmarks used for evaluating FHE-based inference and allow direct comparison with prior work. Input size affects only ciphertext width, not multiplicative depth, so the same encryption parameters apply to higher-resolution datasets. To assess scalability, we trained ResNet-32 on TinyImageNet. A single quadratic model reaches 56.78% accuracy versus 61.37% for the ReLU baseline, and a 2-model ensemble reaches 62.13%. This shows that quadratic training remains stable at larger resolutions and that ensembling compensates for approximation error. Extending to full ImageNet is feasible with increased inference time, and our automated pipeline makes this straightforward. The primary cost is benchmarking encrypted inference at that scale, not model training or correctness. To the best of our knowledge, no prior work has demonstrated LFHE-based private inference on ImageNet-scale inputs, making our findings an important empirical step toward this goal.

---

### Official Review · Reviewer_gvwz · 2025-11-06

**Soundness:** 3
**Presentation:** 3
**Contribution:** 2
**Rating:** 4
**Confidence:** 4

**Summary:**

This paper proposes a technique for optimizing a ResNet model to enable secure AI inference under leveled homomorphic encryption (HE) without requiring bootstrapping. In particular, the authors design a training strategy that allows the ReLU activation function to be replaced with a quadratic polynomial, and they effectively apply an ensemble method to maintain high accuracy on the CIFAR-10 and CIFAR-100 datasets while keeping the activation function as a low-degree polynomial.

**Strengths:**

They reduce the multiplicative depth by combining layers, enabling the implementation of ResNet under leveled homomorphic encryption (LHE). In addition, they introduce a slice ensemble clustering method, which leverages unused SIMD slots to perform parallel computations across multiple models simultaneously, thereby improving training stability.

This is a clear and meaningful contribution, as implementing a full architecture under LHE is by no means an easy task. Moreover, simplifying the model while employing an ensemble-based approach is a valid and effective strategy that could generalize well to other datasets and models—especially since LHE/FHE schemes such as CKKS support SIMD operations, making ensemble-based methods particularly advantageous.

From a machine learning perspective, demonstrating that good performance can still be achieved using a quadratic activation function is significant. The introduction of a novel training method to make this possible is also intriguing. If the proposed approach is validated further, it could provide valuable insights for the privacy-preserving machine learning (PPML) community.

**Weaknesses:**

Implementing ResNet under LHE is certainly a remarkable achievement. However, it is difficult to assess how well the proposed method would generalize to models targeting larger datasets, as the expressiveness of the architecture may be reduced once layers are merged. In this regard, it would be interesting to see how the approach performs on ImageNet. Of course, implementing ImageNet-scale models under LHE is known to be extremely challenging. Nonetheless, even if a fully bootstrapped FHE model or a plaintext simulation were used, it would be valuable to experimentally demonstrate that replacing the activation function with a quadratic function leads to meaningful performance improvements.

When applying the ensemble technique to more complex architectures, the inherent limitations of LHE imply that the ensemble complexity must increase to achieve deeper expressiveness. This, in turn, could reduce efficiency if the number of models exceeds the maximum SIMD slot capacity, as additional inference steps would be required. For instance, it would be helpful to discuss how the proposed ensemble approach might behave on ImageNet-scale tasks—not necessarily by performing an actual homomorphic implementation, but at least by providing a conceptual or experimental discussion of its scalability and expected runtime performance trends.

It seems that the discussion of tower reuse in the paper is based on outdated techniques. The claim that ‘accuracy degrades as the gap between RNS primes and the scaling factor increases’ only applies when the scaling factor is fixed and maintained at its initial value throughout the computation. However, this issue has been fully resolved in the following paper. Please refer to it. It would be advisable to rewrite the section on tower reuse to reflect the implications of this newer method.

Kim, Andrey, Antonis Papadimitriou, and Yuriy Polyakov. "Approximate homomorphic encryption with reduced approximation error." Cryptographers’ Track at the RSA Conference. Cham: Springer International Publishing, 2022.

Moreover, the statement that the security level is proportional to N/Q appears to be inaccurate. Security in homomorphic encryption is not determined by such a simple proportional relationship. For example, please refer to the following paper for a more accurate treatment of this issue. It would also be beneficial to consult a wider range of recent papers on homomorphic encryption to ensure the discussion reflects the current understanding in the field.

Kirshanova, Elena, Chiara Marcolla, and Sergi Rovira. "Guidance for efficient selection of secure parameters for fully homomorphic encryption." International Conference on Cryptology in Africa. Cham: Springer Nature Switzerland, 2024.

**Questions:**

1. Even at the plaintext level, could you verify whether the quadratic activation function can effectively replace the original one when applied to ImageNet?
2. Could you discuss the potential advantages and disadvantages of applying the ensemble method to large-scale datasets such as ImageNet?

---

> ### Author Response · Authors · 2025-12-02
>
> **Impact of Layer Fusion on Model Expressiveness.** Layer fusing in our method preserves exact functional equivalence and therefore does not reduce expressive power. The fused layers are algebraically identical to the original composition of convolution, batch normalization, activation, and residual addition. As shown in Section 3.1, each fusion is simply a symbolic substitution, meaning we rewrite the composed functions into a single closed-form expression without altering their input-output behavior. These transformations are applied after training and do not modify the function class the network can represent. Thus, expressiveness is unchanged; accuracy depends only on the choice of activation, training procedure, and possible ensemble, not on layer fusion. Our contribution is the first to show that extensive algebraic fusion across ResNet blocks can be applied at scale for FHE without altering the representable hypothesis class, which has not been demonstrated in prior PPML works.
>
> **Feasibility of Scaling to ImageNet.** The input size affects only width (number of slots needed), not multiplicative depth, so ImageNet can be evaluated under LFHE using the same encryption parameters and circuit depth. Higher-resolution images simply require more ciphertexts per channel, which increases inference time but does not affect correctness or feasibility. To evaluate whether the quadratic activation scales beyond CIFAR, we trained ResNet-32 on TinyImageNet (200 classes, 64×64 resolution). A single quadratic model achieves 56.78% accuracy, compared to 61.37% for the ReLU baseline, and a 2-model ensemble reaches 62.13%. These results show that the penalty-based quadratic training remains stable at higher resolutions and that ensembling effectively compensates for approximation error. This provides strong empirical evidence that our approach generalizes to larger-scale datasets and supports the feasibility of extending it to full ImageNet. To the best of our knowledge, no prior work has demonstrated LFHE-based private inference on ImageNet-scale inputs, making our findings an important empirical step toward this goal.
>
> **Scalability of the Ensemble Method on Larger Datasets.** Larger inputs inevitably reduce efficiency in FHE because more ciphertexts are required and ensembles may no longer fit within a single packing, increasing latency. This limitation is inherent to all data layouts, since higher-resolution inputs eventually saturate slot capacity. Even without implementing ImageNet-scale models, we can reason about expected trends: inference cost will grow linearly with ciphertext count. Our automated training and conversion pipeline makes scaling to larger datasets straightforward; the practical bottleneck is benchmarking encrypted inference due to the large number of parameters involved. To provide empirical evidence of scalability without a full ImageNet implementation, we are extending experiments to TinyImageNet on ResNet-32, which will clarify runtime behavior as input sizes and model complexity increase.
>
> **Tower Reuse and Its Intended Purpose.** Tower Reuse is not introduced to control approximation error; that effect is only incidental. Its purpose is to reduce the number of RNS towers, which directly accelerates homomorphic operations. The method presented in its generalized form replaces $t$ $B$-bit towers with a single $(tB)$-bit tower. In our implementation, as shown in Table 1 and detailed in Section E.4, the method replaces two 20-bit (or 22-bit) towers with a single 40-bit (or 44-bit) tower, cutting the tower count roughly in half. The smaller approximation error from rescaling simply results from using larger and fewer moduli, and is a byproduct rather than the goal of Tower Reuse.
>
> **Clarification of Security Parameters.** We agree that the original phrasing was inaccurate. What we intended to convey is that, in FHE parameterization, increasing N is associated with higher security, while increasing Q is associated with lower security. We will rephrase the statement to avoid implying any proportional relationship.

---

> > ### Author Response · Authors · 2025-12-02
> >
> > **Answer to Q1:** A degree-2 activation does scale beyond CIFAR. On TinyImageNet (200 classes, 64×64), a single quadratic ResNet-32 reaches 56.78% accuracy versus 61.37% for the ReLU baseline, and a 2-model quadratic ensemble reaches 62.13%. This mirrors the paper’s observation that ensembles compensate for polynomial approximation error and close most of the gap. These results indicate that the quadratic activation remains effective on larger datasets. Expanding to full ImageNet is feasible but will require additional training time. We demonstrate, for the first time, that a ResNet trained end to end with a true degree 2 activation can maintain stability and competitive accuracy at higher resolutions without resorting to high-degree polynomials.
> >
> > **Answer to Q2:** In FHE, activations must be polynomial, so functions like ReLU are replaced with polynomial approximations. Higher-degree polynomial approximations improve accuracy but make inference very slow because they increase multiplicative depth and force many rounds of bootstrapping. Low-degree polynomials are fast but introduce larger approximation errors. An ensemble helps recover this lost accuracy, but on large inputs such as ImageNet the inference cost increases because more ciphertexts are needed to hold multiple model instances. The additional overhead is still much smaller than the cost of using high-degree polynomials. Moreover, techniques such as the parameter clustering described in the paper keep the ensemble overhead manageable. For the fastest possible inference on large inputs, one would avoid the ensemble and adopt a more compact data layout, while utilizing our other methods: degree-2 polynomial activation, structural optimizations, and slice clustering. If accuracy must remain close to the ReLU baseline, ensemble is used at some extra cost in inference time.

---

### Meta-Review · Area_Chair_XRR1 · 2026-01-12

**Summary:**

The overall evaluation leans toward rejection, with three weak rejections and one weak acceptance. Reviewers agree that the paper is technically solid and clearly written, and that it demonstrates a notable engineering achievement: executing relatively deep ResNet models (18/20/32) fully under leveled FHE without bootstrapping, achieving substantial speedups while maintaining competitive accuracy. However, reviewers raised substantial concerns about scope and impact. The evaluation is limited to legacy CNN architectures and small datasets (CIFAR-10/100), with unclear implications for modern large-scale settings. Several reviewers questioned whether the absence of bootstrapping reflects a meaningful advance or simply a constraint tied to bounded depth. Others challenged the novelty of individual techniques relative to prior PPML work, the correctness or currency of certain cryptographic discussions (e.g., security parameterization, tower reuse), and the scalability of ensemble and clustering methods in memory- and slot-constrained regimes. Overall, the paper is seen as a strong, careful optimization within a narrow but still-relevant setting, with impact limited by generality and positioning.

**Reviewer Concerns:**

**The rebuttal was submitted in December, and consequently no reviewers participated in the discussion.**

The authors addressed concerns by explaining that layer fusion preserves functional equivalence and does not reduce the hypothesis class. Scalability questions were partially answered with new TinyImageNet results, demonstrating that quadratic activations and small ensembles remain effective at higher resolution. Misstatements regarding security parameters and the intent of tower reuse were acknowledged and corrected, and comparisons with prior work (e.g., PILLAR, grafting) were sharpened by emphasizing the unique combination of minimal-degree activation, depth reduction, and full LFHE execution. Reviewers’ technical questions on numerical stability, training stability, and ensemble overhead were answered with detailed explanations that improved confidence in correctness.

Despite these clarifications, broader impact concerns persist. The work remains confined to CNNs and ResNet-style architectures, with no concrete pathway shown for extending the techniques to modern models such as Transformers or ViTs. While TinyImageNet results help, full ImageNet-scale validation (even in plaintext) is still missing, leaving questions about real-world relevance. The ensemble approach’s dependence on unused SIMD slots and the memory costs of clustering techniques continue to raise scalability concerns for larger inputs and deeper models. Finally, several reviewers still view many contributions as incremental or compositional relative to existing PPML techniques, limiting the perceived novelty despite strong execution.

**Reviewer Scores:**

**The rebuttal was submitted in December, and consequently no reviewers participated in the discussion.**

Reviewer gvwz: Likely unchanged (Rating: 4). The rebuttal addressed most technical critiques, but broader scalability and impact concerns remain.

Reviewer P459: Likely unchanged (Rating: 4). Core skepticism about relevance beyond legacy CNNs was not fully alleviated.

Reviewer fn4b: Likely unchanged (Rating: 4 ). Clarifications improved confidence but did not fundamentally change novelty concerns.

Reviewer xdM9: Likely unchanged (Rating: 6). Already positive; rebuttal reinforced technical strength but acknowledged limitations.

---

### Decision · Program_Chairs · 2026-01-26

Reject